

# An Optimal Estimation Algorithm for the Retrieval of Fog and Low Cloud Thermodynamic and Micro-physical Properties

Alistair Bell[a], Pauline Martinet[a], Olivier Caumont[a,b], Frédéric Burnet[a], Julien Delanoë[c], Susana Jorquera[c], Yann Seity[a], and Vinciane Unger[a]

[a]CNRM, Université de Toulouse, Météo-France, CNRS, Toulouse, France
[b]Météo-France, Direction des opérations pour la prévision, Toulouse, France
[c]Laboratoire Atmosphères, Milieux, Observations Spatiales/UVSQ/CNRS/UPMC, Guyancourt, France
**Correspondence:** Alistair Bell (alistair.bell@meteo.fr)

**Abstract.** A new generation of cloud radars, with the ability to make observations close to the surface, presents the possibility of observing fog properties with better insight than was previously possible. The use of these instruments as part of an operational observation network could improve the prediction of fog events, something which is still a problem for even high-resolution Numerical Weather Prediction models. However, the retrieval of liquid water

content (LWC) profiles from radar reflectivity alone is an under-determined problem, something which ground-based microwave radiometer observations can help to constrain. In fact, microwave radiometers are not only sensitive to temperature and humidity profiles but also known to be instruments of reference for the liquid water path. By providing the thermodynamic state of the atmosphere, to which the formation and evolution of fog events are highly sensitive, in addition to accurate liquid water path, which can be used to constrain the LWC retrieval from the cloud

radar alone, combining microwave radiometers with cloud radars seems a natural next step to better understand and forecast fog events.

To that end, a newly developed one dimensional variational (1D-Var) algorithm designed for the retrieval of temperature, specific humidity and liquid water content profiles with both cloud radar and microwave radiometer (MWR) observations is presented in this study. The algorithm was developed to evaluate the capability of cloud

radar and MWR to provide accurate LWC profiles in addition to temperature and humidity in view of assimilating the retrieved profiles into a 3D/4D-Var operational assimilation system.

The algorithm is firstly tested on a synthetic dataset, which allows the evaluation of the developed algorithm in idealised conditions. It is then tested with real data from the recent field campaign SOFOG-3D, carried out with the use of LWC measurements made from a tethered balloon platform.

As expected, results from the synthetic dataset study were found to contain lower errors than that found from the retrievals on the dataset of real observations. It was found that retrieval of LWC can be obtained on idealised conditions with an uncertainty of less than $0.04\,\mathrm{g\,m^{-3}}$. With real data, as expected, retrievals with a good correlation (0.7) to in-situ measurements, but with a higher uncertainty than the synthetic dataset, of around $0.06\,\mathrm{g\,m^{-3}}$, was found. This was reduced to $0.05\,\mathrm{g\,m^{-3}}$ when an accurate droplet number concentration could be prescribed to the

algorithm. A sensitivity study was conducted to discuss the impact of different settings used in the 1D-Var algorithm





and the forward operator. Additionally, retrievals of LWC from a real fog event observed during the SOFOG-3D field campaign were found to significantly improve the operational background profiles of the AROME model (Application of Research to Operations at MEsoscale) showing encouraging results for future improvement of the AROME model initial state during fog conditions.

## 1  Introduction

Incorrect fog forecasts have been shown to cause major disruption, especially in the aviation industry (Gultepe et al., 2007). Despite the development of high-resolution numerical weather prediction (NWP) models, the prediction of fog events is still prone to large errors (Steeneveld et al., 2015; Philip et al., 2016). Improving the initial conditions of NWP models through assimilating new observations is one way in which forecasts may be improved (Morss and Emanuel, 2002; Martinet et al., 2020). Technological improvements in ground-based remote sensing instruments present the opportunity to expand the operational observation network in a region of the atmosphere both important for fog prediction and typically under-sampled: the boundary layer.

Measurements of brightness temperatures made by ground-based microwave radiometers (MWR) are sensitive to temperature and humidity profiles, and the total liquid water path (LWP) of the atmosphere. Equally, cloud radars measuring radar reflectivity are sensitive to the liquid water content at different altitudes throughout the atmosphere from as little as 40 m from the surface (Delanoë et al., 2016). Through the combined assimilation of cloud-radar and microwave radiometer observations, it may be possible to improve the initial conditions of temperature, humidity, and liquid water content in NWP models, that could lead to an improved fog prediction. Even if a direct assimilation of raw measurements should be the most optimal, assimilation of retrieved profiles has often been used as a first step towards a direct assimilation (Bauer et al., 2006; Janisková, 2015) and has been chosen in this study. A first step towards the assimilation of MWR and cloud radar observations thus relies on the combined retrievals of temperature, humidity and LWC profiles.

One issue facing the retrieval of temperature, humidity and LWC profiles from MWRs and cloud radars, is that the retrievals are typically under-constrained. That is to say that a multitude of possible atmospheric states could lead to the same observed brightness temperatures and multiple values of LWC with differing droplet size distributions could cause the same observed radar reflectivities. For this reason, retrievals of temperature, humidity or LWC using cloud radars or MWRs typically employ further constraints. This can be done with physical parameterisations- for example, about the size distribution of hydrometeors or adiabaticity- (Fox and Illingworth, 1997; Pospichal et al., 2012), variational retrieval methods which include additional information on an *a priori* estimate of the atmospheric state (Martinet et al., 2015, 2017), instrumental synergy (Matrosov et al., 1992; Crewell and Löhnert, 2003; Tinel et al., 2005) or a combination of these techniques (Löhnert et al., 2007; Che et al., 2016; Ebell et al., 2017; Turner and Löhnert, 2021).





Variational methods constrain retrievals with prior information, which is otherwise known as a background, an *a priori* or a 'first guess'. This consists of the variables that will be retrieved with the algorithm at a specified vertical resolution. The retrieval algorithm makes increments to this background in such a manner that the retrieved profile will become in better agreement with the observations than the background profile. How closely the retrieved profile resembles the background profile and the observations will depend on the respective estimated errors of both sources of information.

Variational algorithms making retrievals of humidity, temperature and LWC using cloud radar-MWR synergy using a static climatological background or RS ascents have already shown promising results (Ebell et al., 2017; Löhnert et al., 2007). Similarly, Martinet et al. (2020) have demonstrated that a variational algorithm using MWR observations in a complex terrain showed that a more accurate background profile resulted in improved retrievals of temperature and humidity profiles.

This study aims at taking advantage of newly developed 95 GHz cloud radars to evaluate how synergistic retrievals of temperature, humidity and LWC profiles combining a MWR with a 95 GHz cloud radar could help to improve the initial state of the AROME convective scale model. The final aim is the assimilation of the retrieved profiles to evaluate how fog forecast could be improved by an improved initial state. However, this study concentrates on the first part of the assimilation strategy by evaluating the capability of a new extended one dimensional variational (1D-Var) algorithm to assimilate both 95 GHz cloud radar and MWR observations in an optimal way.

## 2 Methodology

Variational approaches to solve under-determined problems follow the framework outlined in Rodgers (2000). While more complex variational approaches such as 3D- and 4D-Var aim to combine model and instrumental measurement information throughout three spatial dimensions and one temporal dimension, the 1D-Var algorithm outlined here does so in one vertical spatial dimension.

### 2.1 MWR and cloud radar observations

#### 2.1.1 BASTA Cloud Radar

Cloud radar measurements were provided by the Bistatic Radar System for Atmospheric Sounding (BASTA), which uses a 95 GHz frequency (Delanoë et al., 2016). The instrument employs a frequency modulated-continuous wave (FMCW) transmitter for which the change in frequency is used for the ranging of targets. This is in contrast to the pulsed wave transmitter which is conventionally used, but is required to be more powerful and thus significantly more expensive, making the cost of the instrument restrictively high for wide spread deployment in an observation network.





Cloud radars typically use a frequency in the Ka- (24 GHz to 40 GHz) or W-bands (75 GHz to 110 GHz) of the radar spectrum. The backscatter efficiency of a reflected target will be proportional to the sixth power of the diameter of a target, and the fourth power of the frequency of the transmitted wave whilst inside the Rayleigh scattering regime (McCartney, 1976). Radars using a higher frequency will therefore have a greater sensitivity to targets of a size described by Rayleigh scattering (smaller than around 0.3 mm for 95 GHz radar). It does, however also mean that signal will become attenuated by particles more quickly. Due to the fact that cloud radars are designed for very small targets over a relatively short range, a higher frequency is employed compared to operational precipitation radar.

The BASTA cloud radar also uses a separate transmitter and receiver (bistatic), whereas in other systems, the receiver must be switched off during the transmission of a pulse, and signal from targets close to the instrument can not be detected due to the insufficient transmission time of the wave. The minimum detectable target for the BASTA radar (called radar sensitivity hereafter) is as close as 40 m, though the sensitivity at this range is degraded due to the coupling between the transmitter and receiver. This occurs when signal emitted from the transmitter is collected directly by the receiver (as opposed to being reflected by a target). This can cause large levels of noise and increase the minimum detectable radar reflectivity signal in some gates and render other gates unusable. It was found that the gates including and below 37.5 m were unusable in daylight hours, and gates including and below 25 m were unusable during nighttime hours due to coupling (Jorquera and Delanoë, 2020).

The radar can operate with three different configurations with differing maximum altitudes and range gate resolutions, with the highest resolution being 12.5 m with a vertical range of 12 km and the lowest resolution of 100 m with a vertical range of 18 km. The instrument is also able to combine the different modes to give an increased resolution near the surface and improved sensitivity at larger ranges. In this configuration, as shown by the Contoured Frequency by Altitude Diagram (CFAD) in figure 1, the radar sensitivity is discontinuous due to the decreased vertical resolution from 12.5 m below 500 m to 25 m between 500 m and 5000 m, and 100 m above 5000 m.

The radar sensitivity, $Z_{min}$, above 200 m was found by using the analytical relation described in equation (1), where $r$ is the range gate of the radar observation and $r_0$ is a range at which the sensitivity is known. These sensitivity values were found by fitting equation (1) to the minimum values obtained in and figure 1, an approach which is explained in detail in (Wattrelot et al., 2014).

$$Z_{min}(r) = 20 \cdot \log_{10}(\frac{r}{r_0}) + Z_{min}(r_0) \tag{1}$$

Below 200 m, the coupling between the transmitter and receiver meant that additional noise is present for range gates closer to the ground, and the analytical relation could not be used. Instead, the radar sensitivity was found manually for each range gate below 200 m by a fit to the BASTA CFAD.

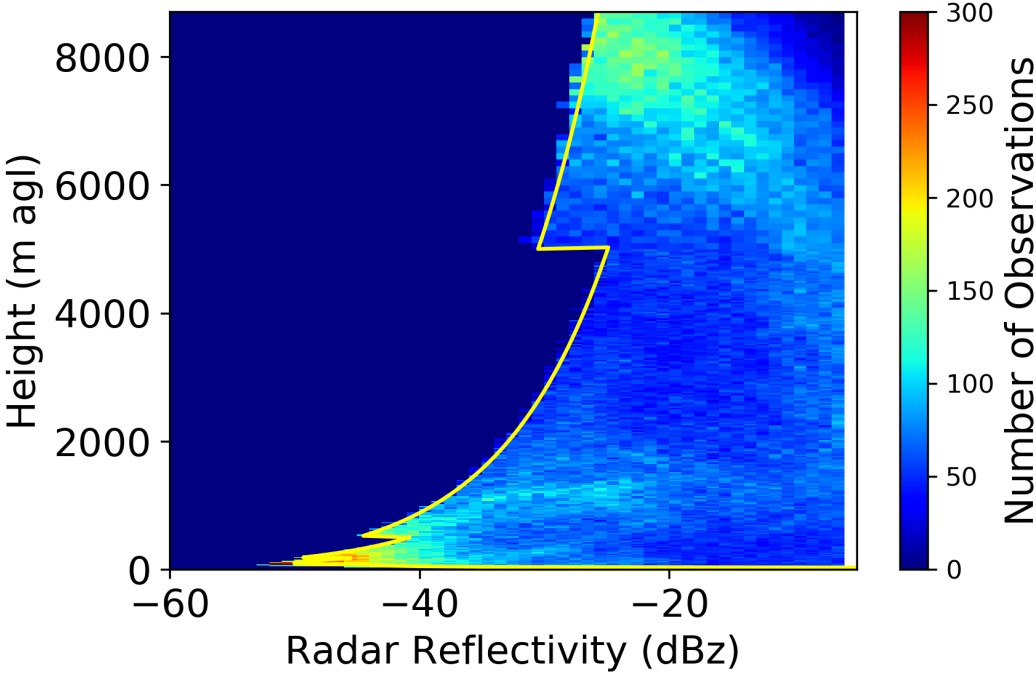

**Figure 1.** Contour frequency by altitude diagram of the number of observations made by the radar for reflectivity intervals (in dBz). The sensitivity of the instrument is marked in the yellow line.

### 2.1.2 The HATPRO Microwave Radiometer

The humidity and temperature profiler (HATPRO) (Rose et al., 2005) is a two-band microwave radiometer with seven channels in each band. The first spectral K-band targets the water vapour absorption line at 22.24 GHZ while the second spectral V-band targets the oxygen complex absorption line at 58 GHz. The radiometer output voltage is directly converted into brightness temperatures via the Planck function. In this study, the third channel of the K-band measuring the downwelling emission of the atmosphere at 23.84 GHz had to be discarded due to a hardware

problem identified during the experiment. Brightness temperatures observed only in the following 13 channels are used in this study: 22.24, 23.04, 25.44, 26.24, 27.84 and 31.4 GHz for the K-band and 51.26, 52.28, 53.86, 54.94, 56.66, 57.3 and 58 GHz for the V-band. The radiometer is also able to scan at elevation angles ranging from 0° (horizontal) to 90° (vertical), to increase the vertical resolution of temperature profiles by assuming spatial homogeneity in the neighborhood of the instrument. For the configurations that were used in this study, elevation scans with 10 angles[1]

from 90° down to 4.2° above the surface were performed once per 10 min, with the radiometer facing vertically at

[1]90°, 30°, 19.2°, 14.4°, 11.4°, 8.4°, 6.6°, 5.4°, 4.8°, 4.2°





other times. When using low elevation angles in this study, only the four most opaque channels (above 54 GHz), ensuring the spatial homogeneity assumption in the vicinity of the instrument are used.

In order to ensure that biases are not present, the radiometer is calibrated before use in a field campaign with the aid of a black body target (with assumed emissivity = 1), which is cooled with liquid nitrogen, which has a well known boiling temperature of 77.5 K. The radiometer also has an internal black body target, which is not heated or cooled, but contains an accurate thermometer so that the black body emission may be accurately estimated.

## 2.2    1D-Var Algorithm

A variational algorithm aims to minimise the departures from the observations and the background profile, weighted by the expected errors of the background profile and the observations. In the case of the MWR and cloud radar, the observed measurements of radar reflectivity and brightness temperatures (BT) are not the same as the variables being retrieved, known as the control variables. This necessitates the need for forward operators to convert the control variables into equivalent brightness temperatures and radar reflectivity. A cost function J, shown in (2), may be calculated, where $\mathbf{x}_b$ is the background state, $\mathbf{B}$ is the background error covariance matrix, $\mathbf{y}$ is the observation vector, $\boldsymbol{F}$ is the forward operator and $\mathbf{R}$ is the observation error covariance matrix. In this study, the observation vector $\mathbf{y}$ contains both the BASTA reflectivity ($\mathbf{Z}$) of the 90 range gates corresponding closest to the heights of the 90 AROME background profile vertical levels and the MWR ($\mathbf{BT}$) observed both at zenith and off-zenith elevation angles. The notation used here is that of Bouttier and Courtier (2002), with the exception that $\boldsymbol{F}$ is used for the forward operator to clearly distinguish it from the Jacobian matrix, $\mathbf{H}$. The ($\mathbf{BT}$) vector thus corresponds to 13 channels at zenith and 4 channels at each of the 9 lower elevation angles (see section 2.1.2), which corresponds to a total size of 49 measurements. At all vertical gates, when the BASTA reflectivity is below the minimum detectable signal (shown in figure 1) the reflectivity is set to this minimum detectable signal.

Radar reflectivity observations are provided with a mask defining the type of hydrometeor which the observation corresponds to (liquid water/airborne plankton, ice, or drizzle/rain). Retrievals are not attempted when rain is present in any pixel of the observations, due to the added complexity for making retrievals when the radome of the radar is wet (due to unknown attenuation effects). Where ice clouds are present in the observations above a liquid water cloud, retrievals of liquid water content in the lower part of the atmosphere are still performed by filtering out the ice cloud signal from the radar reflectivity. To do this, the BASTA reflectivity is set to the radar sensitivity at the range gate containing ice. The aim of the algorithm is to focus on retrievals in warm fog only, but, as the MWR is not affected by the presence of ice, this configuration still allows the additional retrievals of temperature and humidity even during the presence of ice clouds.

In this study, the background state $\mathbf{x}_b$ is defined as the AROME model 90 levels of temperature, specific humidity and liquid water content. Background profiles taken from the AROME model were found to commonly contain a mix of different hydrometeors alongside LWC, most frequently rain or ice. As the algorithm is currently developed only for multi-layer liquid clouds, 1D-Var retrievals are not performed when the AROME background profile contains a





significant amount of ice or rain. To evaluate the significance of the amount of rain and ice in the AROME background profile, a similar method as explained in Bell et al. (2021) has been used. To that end, the radar reflectivity was simulated from the background profiles with all hydrometeors, and then with only LWC. If the profile contained pixels with differences of more than $3\,\mathrm{dB}$ between the simulation with all hydrometeors and with only LWC, the other hydrometeors were considered to make up a significant portion of the cloud, and 1D-Var retrievals are not performed

for this case. In the other cases, 1D-Var retrievals are performed considering rain and ice water contents as negligible. No correction is brought to the AROME LWC background profile when there is a cloud in the observation but not in the background state and vice-versa. In fact, the 1D-Var algorithm should easily remove non-observed cloud layers in the background. This is more complex when the AROME background is clear but the radar observation is not. To deal with this issue, new developments are proposed during the computation of the Jacobian matrix for the cloud

radar reflectivity. This is discussed later on in this section.

     The 1D-Var algorithm aims to minimise the cost function for the control variable state $\mathbf{x}$, for which the statistically optimal state given all the input components is considered to be found.

$$\mathrm{J}(\mathbf{x}) = \frac{1}{2}(\mathbf{x} - \mathbf{x}_{\mathrm{b}})^{\mathrm{T}}\mathbf{B}^{-1}(\mathbf{x} - \mathbf{x}_{\mathrm{b}}) + \frac{1}{2}(\mathbf{y} - \boldsymbol{F}(\mathbf{x}))^{\mathrm{T}}\mathbf{R}^{-1}(\mathbf{y} - \boldsymbol{F}(\mathbf{x})) \tag{2}$$

     The minimisation of the cost function is performed by iteration, with subsequent values of $\mathbf{x}$ being found through

equation (3), where $\mathbf{x}_{i+1}$ is the following state vector, and $\mathbf{x}_0$, the initial state, is equal to $\mathbf{x}_b$. Factor $\gamma$ is a coefficient specified by the Levenberg–Marquardt descent algorithm. Here $\mathbf{H}_i$ represents the Jacobian matrix, the predicted sensitivity of the observation matrix to a change in state $\mathbf{x}_i$ ($\mathbf{H_i} = \partial \boldsymbol{F}(\mathbf{x_i})/\partial \mathbf{x_i}$).

$$\mathbf{x}_{i+1} = \mathbf{x}_i + \left((1+\gamma)\mathbf{B}^{-1} + \mathbf{H}_i^T \mathbf{R}^{-1} \mathbf{H}_i\right)\left(\mathbf{H}_i^T \mathbf{R}^{-1}(\mathbf{y} - \boldsymbol{F}(\mathbf{x}_i)) - \mathbf{B}^{-1}(\mathbf{x}_i - \mathbf{x}_b)\right) \tag{3}$$

     For this experiment, the 1D-Var package maintained by the NWP Satellite Application Facility (NWPSAF;

https://www.nwpsaf.eu/site/software/1d-var/, last access: 25 October 2021), modified for the 1D assimilation of ground-based MWR observations (Martinet et al., 2020), has been extended to cloud radar observations. The forward operators $\boldsymbol{F}$ used both for MWR and cloud radar observations are described in section 2.3.

     The Jacobian matrix for cloud radar reflectivity was calculated through the brute force method, which involved running the forward models with small perturbations to each element of $\mathbf{x}_i$ (De Angelis et al., 2016), due to the

difficulties of linearising the forward operator. This method has been used in previous studies investigating the data assimilation of radar reflectivity, and been shown to be an effective way to estimate the Jacobian matrix (Thomas et al., 2020). This is formulated in equation (4), where $\delta\mathbf{x_k}$ is the perturbation made to element $k$ of the current state





vector $\mathbf{x_i}$ for a state vector of size $\mathbf{n}$, and $\boldsymbol{F_K}(\mathbf{x})$ is the simulated observation corresponding to the $\mathrm{K_{th}}$ observation from state $\mathbf{x_i}$, for an observation matrix of size $\mathbf{N}$.

$$195 \quad \mathbf{H} = \frac{\partial \boldsymbol{F}(\mathbf{x})}{\partial \mathbf{x}} \approx \begin{bmatrix} \frac{\boldsymbol{F_1}(\mathbf{x}+\delta\mathbf{x_1})-\boldsymbol{F_1}(\mathbf{x})}{\delta\mathbf{x_1}} & \frac{\boldsymbol{F_1}(\mathbf{x}+\delta\mathbf{x_2})-\boldsymbol{F_1}(\mathbf{x})}{\delta\mathbf{x_2}} & \cdots & \frac{\boldsymbol{F_1}(\mathbf{x}+\delta\mathbf{x_n})-\boldsymbol{F_1}(\mathbf{x})}{\delta\mathbf{x_n})} \\ \frac{\boldsymbol{F_2}(\mathbf{x}+\delta\mathbf{x_1})-\boldsymbol{F_2}(\mathbf{x})}{\delta\mathbf{x_1}} & \frac{\boldsymbol{F_2}(\mathbf{x}+\delta\mathbf{x_2})-\boldsymbol{F_2}(\mathbf{x})}{\delta\mathbf{x_2}} & \cdots & \frac{\boldsymbol{F_2}(\mathbf{x_n}+\delta\mathbf{x_n})-\boldsymbol{F_2}(\mathbf{x})}{\delta\mathbf{x_n}} \\ \vdots & \vdots & \ddots & \vdots \\ \frac{\boldsymbol{F_N}(\mathbf{x}+\delta\mathbf{x_1})-\boldsymbol{F_N}(\mathbf{x})}{\delta\mathbf{x_1}} & \frac{\boldsymbol{F_N}(\mathbf{x}+\delta\mathbf{x_2})-\boldsymbol{F_N}(\mathbf{x})}{\delta\mathbf{x_2}} & \cdots & \frac{\boldsymbol{F_N}(\mathbf{x}+\delta\mathbf{x_n})-\boldsymbol{F_N}(\mathbf{x})}{\delta\mathbf{x_n}} \end{bmatrix} \quad (4)$$

The perturbation size of the forward operator with respect to the different variables has been selected according to the observed linear behavior of the forward operator after testing different values of perturbation with changes of this size.

As previously stated, when the AROME background is clear whereas the BASTA observation shows a cloud layer, the 1D-Var algorithm will not be able to add a new cloud layer due to zero values in the Jacobian matrix. In fact, where the AROME background is clear, the simulated radar reflectivity is below the minimum detectable signal and set to the radar sensitivity to be consistent with the BASTA observations. In that case, a small perturbation in the initial LWC profile is unable to create a radar reflectivity above the radar sensitivity, the difference between the two simulated reflectivities (with the perturbed profile minus the initial profile) will be zero. Consequently, the Jacobian values calculated from the brute force method will be equal to zero, which means that the inclusion of a new cloud layer in an initially dry background profile would not be possible.

Here this work differs from others (Thomas et al., 2020), and Jacobian values are forced to be non-zero when the background profile is clear to give more flexibility to the algorithm to create a cloud layer when necessary. In order to do this, for each range gate, the minimum value of LWC leading to a radar reflectivity value equivalent to the radar sensitivity at that specific vertical altitude has been defined. Jacobians values are then calculated in the neighborhood of this defined minimum LWC values to fill in the Jacobian matrix where the background is clear.

For MWR brightness temperatures, the adjoint of the tangent linear of the fast radiative transfer model RTTOV-gb is used (De Angelis et al., 2016).

The algorithm will converge when a state $\mathbf{x_a}$ is obtained that minimises the cost function. The errors associated with this state are specified by the analysis error covariance matrix, $\mathbf{A}$ (following again the notation of Bouttier and Courtier (2002), and not to be confused with the averaging kernel). This may be found from equation 5, with diagonal terms giving an estimate of the variance of the retrieval error.

$$\mathbf{A} = (\mathbf{H}^T \mathbf{R}^{-1} \mathbf{H} + \mathbf{B}^{-1})^{-1} \quad (5)$$



### 2.3 Forward Models: Radiative Transfer Model and Radar Simulator

Forward models, otherwise referred to as observation operators, are used to convert the control variables of the algorithm into observation type variables, the use of which are a common approach for the assimilation of observations indirectly related to the control variables (Courtier et al., 1998).

     The fast radiative transfer model RTTOV-gb, (Saunders et al., 2018; Cimini et al., 2019) was used as the forward model for brightness temperatures in this experiment. This gives simulated brightness temperatures at the required

microwave frequencies by using a radiative transfer equation for upward looking passive sensors. The model computes the Planck radiances emitted from the top of the atmosphere to the surface, and takes into account the absorption between the height level of the emitted radiance and the surface. In RTTOV-gb, liquid water is taken into account as an absorbing species, meaning that the effects of clouds on observed microwave brightness temperatures may be taken into account (De Angelis et al., 2016). This is necessary in order to more accurately model the brightness

temperatures, which is essential to retrieving accurate profiles of temperature and humidity with this methodology. It also permits the retrieval of the LWP. While several radiative transfer models are able to simulate downwelling radiance with the aforementioned capabilities, RTTOV is designed to make fast calculations, and is thus highly suited to operational variational methods.

     In order to simulate the radar reflectivity from the control variables, a radar simulator for vertically pointing W-

band radar designed by Borderies et al. (2018) was used. Inputs of pressure, temperature, humidity and mixing ratio of five hydrometeor types (liquid cloud, ice cloud, rain, snow and graupel) must be specified. Radar reflectivity is computed at the same resolution as the input profiles, taking into account backscattering due to the five hydrometeor types and attenuation from moist and dry air. The simulator assumes a modified gamma distribution for the size distribution of hydrometeors, consistant with the ICE3 microphysical scheme in the AROME model, with the

distribution coefficients being specified as additional inputs. An evaluation of the radar simulator capability for ground-based 95 GHz cloud radar and sources of uncertainty can be found in Bell et al. (2021).

### 2.4 Background Profile: The AROME model

The background profile is used to constrain the retrieval, and hence the more accurate the background profile is, the more accurate the retrievals are likely to be. The French convection-scale NWP model AROME was hence used

to provide background profiles for this study. The AROME model has 90 vertical levels from the surface to a height of approximately 30 km, and a horizontal resolution of 1.3 km over the domain centered over mainland France, and covering most of western Europe (Brousseau et al., 2011).

     Bell et al. (2021) have shown that large errors due to the spatial and temporal displacement of fog events could be expected from model short-term forecasts during fog events, leading to sub-optimal background profiles for future

1D-Var retrievals. An approach to select a more adapted background profile with reference to the observed radar reflectivity within a 27 km sub-domain and within a 6 hour time window, named the most resembling profile (MRP)





method was therefore proposed, which was shown to significantly reduce innovation (observation minus background) error statistics. This study follows the work of Bell et al. (2021) by using the MRP method to provide an adequate background profile to the 1D-Var retrieval algorithm. The impact of using the MRP method will be discussed in
section 4.3.

## 2.5    Estimation of the Observation Error and Background Error Covariance Matrices

The specification of the errors associated with the background profile and observations instruct the algorithm on the level of trust placed in both the observations and the background profile. The retrieved profile will take into account the relative weight of the observation errors compared to the background errors, thus the smaller either the
background or observation errors are, the more closely the retrieved profile is likely to agree with the background or the observations respectively. It is the role of the background error co-variance matrix to specify the background errors to the 1D-Var algorithm. Background errors are not assumed to be independent of one another, thus co-variances between the different control variables are specified in the off-diagonal terms of this matrix.

There are two common approaches to the modelling of background error statistics: to compute innovation statistics
by comparing background profiles to observations with a low observation error, or to use a surrogate method, which it is assumed can approximate the errors (Fisher, 2003). Due to the difficulty of gathering observations representative for all forecast conditions, and for 3D/4D-Var algorithms, the difficulty of modelling the relation of errors in different locations, surrogate techniques have been a common choice. These surrogate techniques normally involve comparing forecasts run with different lead times (Fisher, 2003; Derber and Bouttier, 1999; Descombes
et al., 2015; Bannister, 2008). The background error covariance matrix used operationally in the AROME 3D-Var is computed from a climatological dataset of 3 hour ensemble forecasts derived from the operational AROME Ensemble Data Assimilation system. Ménétrier and Montmerle (2011) have demonstrated that a $\mathbf{B}$ matrix specifically adapted for fog exhibited significant differences compared to a $\mathbf{B}$ matrix designed for cloud-free areas, and that the use of this led to an improved analysis during fog conditions. The $\mathbf{B}$ matrix used in this experiment was generated in a
similar way applying a fog-mask in order to better represent background-error-covariances specific to fog conditions. More explanations about the $\mathbf{B}$ matrix computation can be found in Martinet et al. (2020).

The observation error covariance matrix ($\mathbf{R}$ matrix) is comprised of the instrumental error for each measurement plus the error contained in the forward operator used to simulate the measurement from the state vector $\mathbf{x}$. For observations made with the microwave radiometer, forward model errors vary significantly between each channel. Uncertainty in the modelling of brightness temperatures resulting from uncertainties in absorption modelling has been assessed to range from 0.3 K at 22.24 GHz up to 3.18 K at 52.24 GHz in winter in the midlatitudes (Cimini et al., 2018). The total uncertainty due to the MWR calibration technique has been assessed to be between 0.2 K and 1.2 K by Maschwitz et al. (2013). Finally, instrumental noise is below 0.5 K at all channels (Rose et al., 2005).





Each source of uncertainty has been added in quadrature to provide the total observation error:

$$\sigma_{tot} = \sqrt{\sigma_{noise}{}^2 + \sigma_{calib}{}^2 + \sigma_{FM}{}^2}$$

with $\sigma_{tot}$ the total observation error, $\sigma_{noise}$ the uncertainty due to the instrumental noise, $\sigma_{calib}$ calibration uncertainties and $\sigma_{FM}$ the uncertainty due to spectroscopic errors in the radiative transfer model. The observation error covariance matrix is assumed to be diagonal with observation error values for each channel provided in table 1.

**Table 1.** Observation uncertainties (K) corresponding to MWR brightness temperature measurements prescribed in the observation-error-covariance matrix for each channel.

| Frequency (GHz): | 22.24 | 23.04 | 25.44 | 26.24 | 27.84 | 31.4 | 51.26 | 52.28 | 53.86 | 54.94 | 56.66 | 57.3 | 58 |
|---|---|---|---|---|---|---|---|---|---|---|---|---|---|
| $\sigma_o$ (K): | 1.34 | 1.71 | 1.08 | 1.25 | 1.17 | 1.19 | 3.21 | 3.29 | 1.30 | 0.37 | 0.42 | 0.42 | 0.36 |

For the cloud radar errors, instrumental error of 2 dB was assumed from work on the calibration of the BASTA cloud radar (Toledo et al., 2020). As discussed in Bell et al. (2021), the primary component of forward model errors in the radar simulator comes from the hypothesis made on the assumed cloud droplet size distribution and was found to be approximately of 3 dB (Bell et al., 2021). A forward model error of 3 dB is thus assumed hereafter. The total expected variance is found from the variance of measurement errors plus the variance of forward model errors, and it follows that the standard deviation, what is considered as the instrumental error here, is the square root of this. The total instrumental error used for the retrievals was therefore considered to be 3.6 dB.

Despite the probability of a degree of correlation between the radar observation errors at different range gates, which could come from both the calibration uncertainties and correlations in the size distribution errors, the **R** matrix was assumed to be a diagonal matrix, i.e. there was assumed to be no correlation in the observation errors. Despite this, tests were made with use of a square **R** matrix with high error correlations between the radar observations, which suggested that little impact would be brought to retrievals with the use of this.

## 3  1D-Var retrieval validation on synthetic dataset

In order to ensure the good behaviour of the newly developed algorithm, a method of verification is needed. Though this could indeed be performed by using a dataset of in-situ real measurements, these will themselves incur a certain amount of instrumental error, and large datasets of temperature, humidity and liquid water content profiles collocated with the BASTA cloud radar and the HATPRO microwave radiometer are rare or non-existent. For this reason, an artificial synthetic dataset of cloud radar reflectivity, microwave radiometer brightness temperatures and background profiles simulated from assumed true profiles of temperature, humidity and liquid water content may be created. Through the validation between 1D-Var retrievals, made using the simulated background profiles and observations, and the true profiles, the algorithm may be verified as working as intended, and the expected improvement made to the background profiles by the retrievals can be quantified. This methodology, commonly used to evaluate the





benefit of new observations (Martinet et al., 2013; Ebell et al., 2017), provides the expected retrieval accuracy under idealized optimal conditions, i.e. without instrumental biases and that forward model assumptions are consistent with the observations. This approach is also valuable to evaluate the sensitivity of the retrievals to the algorithm
settings.

## 3.1   Synthetic Dataset

The requirements for making a synthetic database are that we have a dataset which best resembles that of the real observations. For this, it was required that: i) the true profiles would be physically consistent and resemble atmospheric profiles that would be observed ii) background profiles contain the expected error of the real background
profiles (i.e. representative of the AROME model short-term forecasts) iii) observations contain the expected errors seen in the real observations, plus the expected errors due to forward operator approximations.

To that end, the considered true profiles were generated from the 10 minute to 180 minute AROME model short-term forecasts at the SIRTA observatory (Site Instrumental de Recherche par Télédétection Atmosphérique) (Haeffelin et al., 2005) between November 2018 and February 2019 (Bell et al., 2021). The background profiles were
derived from the true profiles by perturbing the temperature, humidity and LWC profiles according to the expected AROME background error covariance matrix during fog conditions. This was done with the use of the *random.multivariate_normal* function in the python Numpy package (Harris et al., 2020), to produce random vectors having a covariance equal to the **B** matrix. By adding these vectors to the true profiles, a dataset containing the error expected in the real background profiles was created. To ensure that this would reflect the conditions in
which retrievals would be made, a specific fog **B** matrix was computed from one AROME EDA cycle valid for one fog case simulated in November 2018. Similarly, the synthetic observation dataset could be derived by simulating the MWR brightness temperatures and then perturbing these by the expected observation error covariance matrix described in section 2.5. An estimation of radar reflectivity error of 3 dB was made initially and used in the synthetic data study. This is in line with the 3.6 dB estimated uncertainty from Toledo et al. (2020), that was later used in
the application to real measurements.

From the synthetic database, forecasts involving mixed phase cloud were excluded, as the retrieval algorithm has not yet been developed to make retrievals of ice, and where there is radar reflectivity which comes from a mixture of ice and liquid water, it is not currently possible to distinguish between the radar reflectivity signal from ice crystals and the signal from water droplets. In total, 1063 suitable profiles were found involving a range of different synoptic
conditions, with radiative fogs, low stratus cases, and stratus lowering fog found to make up most of the dataset.

## 3.2   Quantification of Retrieval Accuracy

In this section, 1D-Var retrievals have been carried out on the database of suitable liquid cloud profiles. A requirement for the convergence of the algorithm is made based on the value of the cost function, the normalised gradient of the



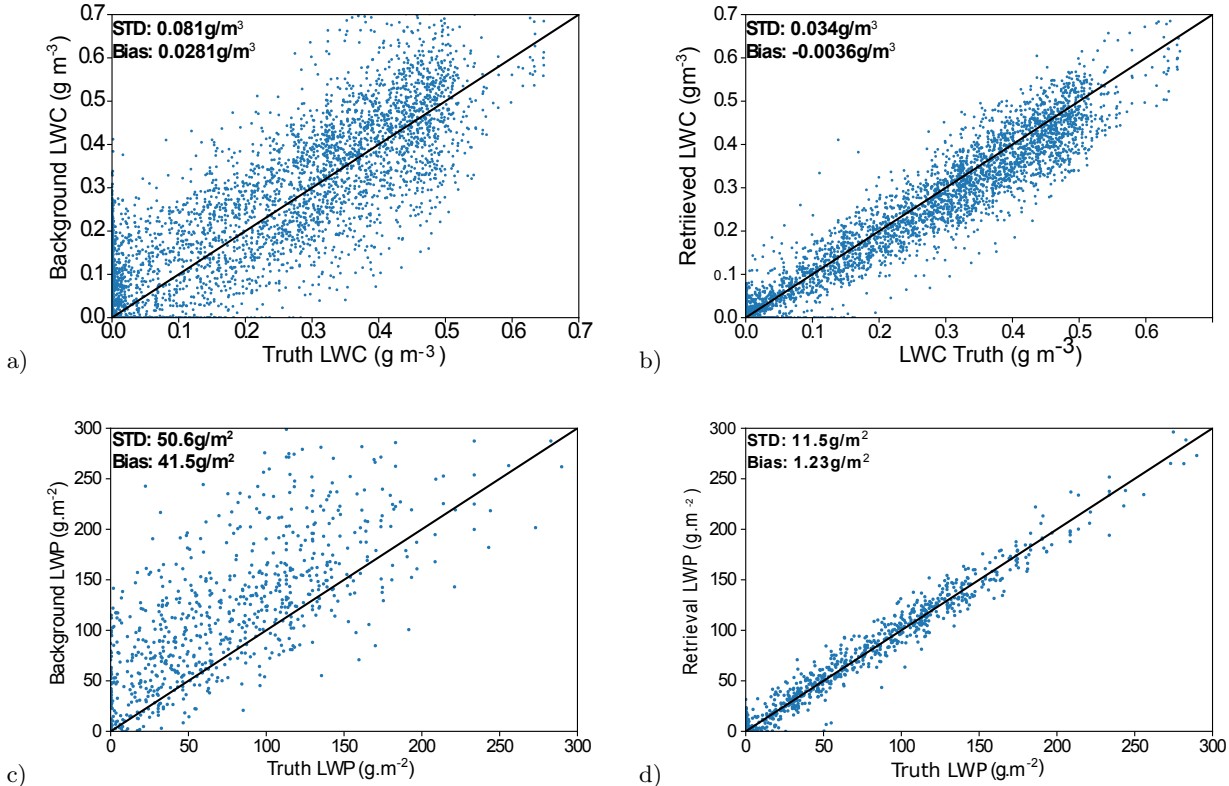

**Figure 2.** Scatter plots showing the values of LWC of the a) background and truth and b) the retrieval and truth; and the LWP for c) background and truth and d) retrieval and truth. Where points lie on the black line, a perfect prediction of the LWC/LWP is made, and the further from the black line, the worse the prediction is. The standard deviation and bias of the background/retrieval - true values is also indicated.

cost function and the gamma factor of the Levenberg–Marquardt minimisation. Successful retrievals were made for
97 percent of the profiles, with a maximum number of iteration set at 15.

Figure 2 shows that the background profiles present a positive bias in the LWC field. This is caused by the way in which perturbations are made to all the fields. By adding or subtracting LWC amounts from the true profiles based on the background error covariances according to a Gaussian distribution, occasionally perturbations will be made that decrease the LWC field to be below zero. As this is un-physical, any values of LWC below zero were set
to zero. As this meant that a net increase was being made to the LWC field, a positive bias was seen. Figure 2 shows that the 1D-Var is able to correct this bias, reducing it from $0.028\,\mathrm{g\,m^{-3}}$ in the background to $0.004\,\mathrm{g\,m^{-3}}$ in the retrieval. A significant improvement is also observed in the total root-mean-square-error (RMSE) from $0.047\,\mathrm{g\,m^{-3}}$

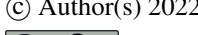



in the background profile to $0.018\,\mathrm{g\,m^{-3}}$ in the retrievals. The correlation with the true LWC values is also improved from 0.90 to 0.98.

As anticipated, the retrieval was also able to improve the values of LWP. From figure 2, the effect of the positive bias on the LWC can be seen more clearly. In fact, in the background profiles, a large positive bias of $41.5\,\mathrm{g\,m^{-2}}$ can be seen, with a standard deviation of LWP errors of $50.6\,\mathrm{g\,m^{-2}}$. The standard deviation of errors is improved by around $75\,\%$ in the retrievals, to reach the values of $11.5\,\mathrm{g\,m^{-2}}$. It can be noted that this estimated uncertainty is much smaller than the expected $20\,\mathrm{g\,m^{-2}}$ when MWR are used alone to derive the LWP (Crewell and Löhnert,
2003) highlighting the potential benefit of the instrumental synergy.

     Another point of interest in the study is to quantify the benefit of the dual retrieval method compared to retrievals made with one instrument alone. As highlighted in section 1, the microwave radiometer is sensitive to the temperature and humidity profiles, but only the integrated value of LWC (the LWP). The radar, meanwhile, is sensitive to the LWC at each range gate observed, but has very little sensitivity to the temperature and humidity. Because of this,
it was not expected that retrievals made with radar observations alone would result in changed temperature and humidity values. However, if appropriate cross-correlations between variables are used during the 1D-Var algorithm, we can expect that an improvement in LWC increments could positively impact the temperature and humidity increments. Additionally, by better locating the cloud in the vertical, we should also improve the radiative transfer simulation of MWR channels sensitive to humidity and LWC during the 1D-var minimisation. If dual instrumental
retrievals were statistically found to be better than retrievals with the radar alone, this would suggest the benefit of a single multi-instrumental algorithm compared to separate algorithms.

     The statistics of the retrieved vertical profiles presented in figure 3 confirm the previous conclusion concerning an initial LWC positive bias in the background profile, with a peak of $0.03\,\mathrm{g\,m^{-3}}$ bias at $500\,\mathrm{m}$. This bias is fairly well corrected for in the retrievals with absolute biases smaller than $0.01\,\mathrm{g\,m^{-3}}$. The standard deviation of
background/retrieval minus truth statistics also show that the most benefit is brought to the retrievals of liquid water content when both instruments are used. It can be noted that the cloud radar shows the largest benefit of either instrument alone but the additional MWR information manages to decrease the standard deviation with respect to truth profiles at all vertical gates with the largest improvement above $1\,\mathrm{km}$. An overall accuracy of $0.02\,\mathrm{gm^{-3}}$ above $400\,\mathrm{m}$ and $0.04\,\mathrm{gm^{-3}}$ below is expected when both MWR and cloud radar measurements are used, giving a
relative standard deviation of errors of around $20\,\%$. When the MWR alone is used the expected retrieval error is approximately twice the one obtained with the synergistic retrieval but already managed to decrease the background error from $0.09\,\mathrm{g\,m^{-3}}$ to $0.05\,\mathrm{g\,m^{-3}}$ at $250\,\mathrm{m}$.

     The effect of the assumed instrumental error was also investigated, with retrievals being made with a cloud radar instrumental error of $1\,\mathrm{dB}$, $3\,\mathrm{dB}$ and $6\,\mathrm{dB}$, whilst the same synthetic database of observations and background profiles
was used. There were no significant differences seen by reducing the observation error from $3\,\mathrm{dB}$ to $1\,\mathrm{dB}$, however, when the instrumental error was increased to $6\,\mathrm{dB}$, the retrieval accuracy was degraded slightly by $0.01\,\mathrm{g\,m^{-3}}$, as shown in table 2.



**Figure 3.** Statistics showing a) the bias and b) the standard deviation of the background/retrieved profiles minus truth for LWC; c) the bias and d) standard deviation of errors for temperature; e) and f) again the same statistics for specific humidity. Retrievals made with only microwave radiometer observations, only radar observations, and both radar and microwave radiometer observations are shown.





**Table 2.** Bias and standard deviation of LWC errors resulting from cloud radar observation errors of different magnitudes being used in the retrieval taking into account all radar vertical bins.

| Observation Errors | STD (g m$^{-3}$) | Bias (g m$^{-3}$) |
|---|---|---|
| 1 dB | $2.9 \times 10^{-2}$ | $0.3 \times 10^{-2}$ |
| 3 dB | $3.4 \times 10^{-2}$ | $0.3 \times 10^{-2}$ |
| 6 dB | $4.7 \times 10^{-2}$ | $0.3 \times 10^{-2}$ |

Figure 3 also shows the statistics for the temperature and humidity fields. As expected, the use of MWR observations significantly decreases the temperature errors with respect to the truth profiles from approximately 1.3
K in the background to 0.7 K in the retrieval at 200 m altitude. At 200 m agl, the standard deviation of errors is slightly improved for the dual retrieval compared to the microwave only retrieval, though these differences can not be considered to be significant.

It can be noted that the radar alone lacks any meaningful sensitivity to either variable, shown in figure 3 c), d), e) and f). Though attenuation of radar reflectivity is affected by the temperature and humidity of the air below the
backscattering target, the perturbation of humidity and temperature from the background profile needed to bring about a change in radar reflectivity, relative to their respective errors, is far higher than the relative change in LWC needed to bring about a change in reflectivity relative to LWC background errors. However, changes to the LWC field can impact the retrieval of temperature and humidity. In fact, as the **B** matrix contains the cross correlation of errors between all retrieved quantities (i.e. temperature, humidity and LWC for 90 height levels, which are the same
as those in the AROME model), a larger perturbation in the LWC field at a certain height could be associated with a larger perturbation in temperature or humidity, should the correlation between the errors of the two variables be strong. When the radar is used in conjunction with the microwave radiometer, the effect of this is generally seen to a greater extent.

Such an impact is seen in the humidity retrievals in figure 3 e) and f) with there being little difference between the
standard deviation of errors in the radar only retrieval compared to the background profile. The standard deviation of errors did, however, become slightly reduced in the dual retrieval configuration compared to the microwave only retrieval. However, for the altitudes where this improvement is seen, there persists a negative bias for the dual radar plus microwave radiometer retrievals which is larger than for the microwave only retrievals.

Due to the existing positive bias in the LWC background profiles, as the retrievals using the cloud radar involve
a net reduction in the LWC, the positive cross correlation of background errors between LWC and specific humidity in this region (for variables at the same height level), could encourage too large reductions in the specific humidity field and the appearance of this negative bias in the humidity retrievals.

A similar impact is observed in figure 3 for temperature retrievals. In fact, the only height at which the standard deviation of temperature retrieval errors with cloud radar alone are visibly different from the standard deviation





of background errors is at 200 m agl, the height at which the largest errors in the LWC field are observed, and the height at which the radar alone retrieval bias is the largest. It should be noted that the bias in the temperature retrievals at this height increases to four times the value of bias in the dual retrieval.

### 3.3 Degrees of Freedom for Signal

It can be useful to know how many independent pieces of information are used in the retrievals. Modern microwave
radiometers contain many channels, between which there is often a high degree of correlation. The number of independent observations is therefore considerably lower than the number of channels. The measurement error will also mean that the information content from the observations will also decrease, meaning that the number of independent pieces of information will be lower than would be found from an observation with zero instrumental error. In Rodgers (2000), this problem is formalised by relating the number of independent columns of information
to the concept of degrees of freedom. We may then ask how many of the degrees of freedom of measurement are related to noise, and how many are related to signal.

To calculate the number of independent pieces of information that are used in the 1D-Var retrieval, formula 6 may be used:

$$\mathrm{DFS} = \mathrm{tr}(\mathbf{I} - \mathbf{A}\mathbf{B}^{-1}) \tag{6}$$

Here, $\mathbf{I}$ is the identity matrix, tr refers to the trace of a matrix and $\mathbf{A}$ is the analysis error covariance matrix described earlier. We may also split the degrees of freedom for signal (DFS) into the three variables being retrieved, showing the DFS for temperature, humidity and liquid water content independently. The mean DFS for all the synthetic profile retrievals made is shown in table 3.

For the liquid water content retrievals, we expect the radar to contribute the most to degrees of freedom for signal
compared to the microwave radiometer measurements. As the DFS coming from a radar is likely increased as the number of range gates affected by a cloud layer increases, the relative DFS for the liquid water content retrievals is calculated instead of the absolute DFS. The relative DFS for LWC is the DFS for LWC divided by the total number of levels containing non-zero LWC values. While MWR provides very few independent pieces of information on the LWC profile with a relative DFS of 2.7 %, a relative DFS of 37 % is obtained by the single use of the cloud radar.
The synergistic use of both MWR and cloud radar manages to increase the relative DFS up to 38.2 %.

When only MWR observations are used and with respective values of 2.31 and 1.75 the DFS for temperature and humidity was smaller than that which has been found in other works (Löhnert and Maier, 2012; Ebell et al., 2017). However, these studies used climatological background profiles which typically have larger background errors. The DFS measures the sensitivity of the retrieved profiled to changes in the true profile. The larger the background errors
are assumed to be, the more weight will be given to the observations, which will change with respect to the true profile while the background remains fixed. Therefore, the more relative weight given to the observations, the higher





**Table 3.** Average DFS for temperature, humidity and LWC retrievals.

| - | Temperature | Specific Humidity | Liquid Water Content (%) |
|---|---|---|---|
| Dual Retrieval | 1.99 | 0.86 | 38.2 |
| Radar Only | 0.11 | 0.09 | 37.0 |
| MWR Only | 2.31 | 0.75 | 2.7 |

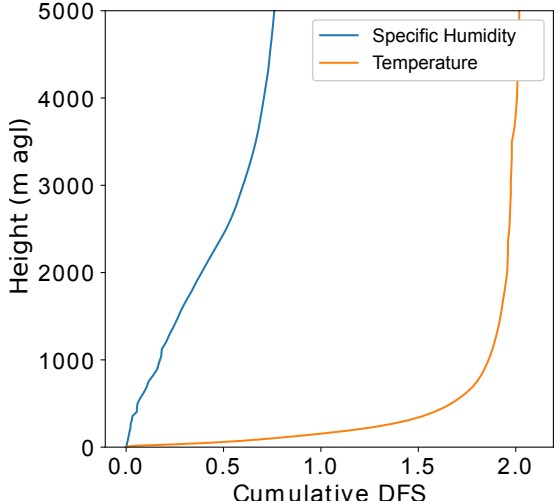

**Figure 4.** The cumulative degrees of signal for freedom for the synthetic temperature (orange line) and humidity (blue line) retrievals.

the DFS will be. As a dynamic background taken from an NWP model is used in this study, the background errors are considerably lower than those of a climatological background approximately by a factor 10 for temperature and factor 5 for humidity, resulting in a lower DFS similarly to what has been observed in Turner and Löhnert (2021). It may be noted that as expected almost no information can be extracted about temperature and humidity from the radar alone. However, through the background error covariance matrix correlations, the synergistic retrievals slightly decrease the temperature DFS while increasing the humidity DFS.

The cumulative DFS may also be examined. This shows at which altitudes the signal used in the retrievals comes from, and is shown in figure 4. At a given altitude, the higher the rate of change of cumulative DFS, the greater the signal used in the retrievals. It may be seen that most information from the temperature retrievals is found in the lowest 750 m, whereas for humidity, it is found mainly between 1000 m to 3000 m.



**Table 4.** Table of instruments that were deployed during the SOFOG-3D field campaign that were used in this study. *The measurement uncertainty is not well defined and may change as a function of the droplet sizes and diameters.

| Instrument Name | Measured Variable | Units | Measurement Uncertainty | Measurement Range | IOP Only |
|---|---|---|---|---|---|
| BASTA Cloud Radar | Radar Reflectivity | dBZ | 2dB | $-52\,\text{dBZ}$ to $20\,\text{dBZ}$ | No |
| PWD22 Visibility Sensor | Meteorological Optical Range | m | 10 % (below 10 km) | 0.01 km to 20 km | No |
| RS-41 Radiosonde | Temperature | °C | 0.3 °C | $-65\,°\text{C}$ to $70\,°\text{C}$ | Yes |
|  | Relative Humidity | % | 4 % | 0 % to 100 % |  |
| HATPRO Microwave Radiometer | Brightness Temperature | K | 0.3 K to 0.5 K | N/A | No |
| Cloud Droplet Probe | LWC | $\frac{\text{g.m}^{-3}}{\text{cm}^{-3}}$ | $\frac{30\,\% \; *}{20\,\%}$ | N/A | Yes |
|  | Number Concentration |  |  |  |  |
| CT25K Ceilometer | Cloud Base Height | m | 2 %+7.5 m | 0 km to 7.5 km | No |

## 4  Application to Real Data

Although the synthetic data study allowed the potential of the algorithm to be shown, it relied on idealised assumptions about instrumental and background errors, which are likely to themselves contain errors in real world applications. To be sure that these assumptions were valid, and to analyse the performance of the algorithm in an operational context, it is necessary to test the algorithm in real-world conditions.

### 4.1  SOFOG-3D Field Campaign

The south-west fog 3D experiment for process studies (SOFOG-3D) field campaign took place between October 2019 and March 2020, and was an observational field campaign focused on fog (Burnet et al., 2020; Martinet et al., 2020). In addition to numerous other measurements across the south-west of France, a main measurement site was located near to the village of Saint-Symphorien in the department of Gironde, 50 km south of Bordeaux. Here, the BASTA cloud radar and HATPRO microwave radiometer were deployed, making continuous measurements throughout the 6-month period. During fog episodes, intensive observation periods (IOPs) took place. During these IOPs, a tethered balloon to which a cloud droplet probe (CDP), optical particle counter (OPC) and cloud condensation nuclei counter (CCNC) were also used for vertical profiling of both aerosol and cloud microphysical properties. During the IOPs, radiosondes recording temperature, humidity and wind were also launched two to four times per night. Table 4 notes the instruments at the supersite which were used in this study.

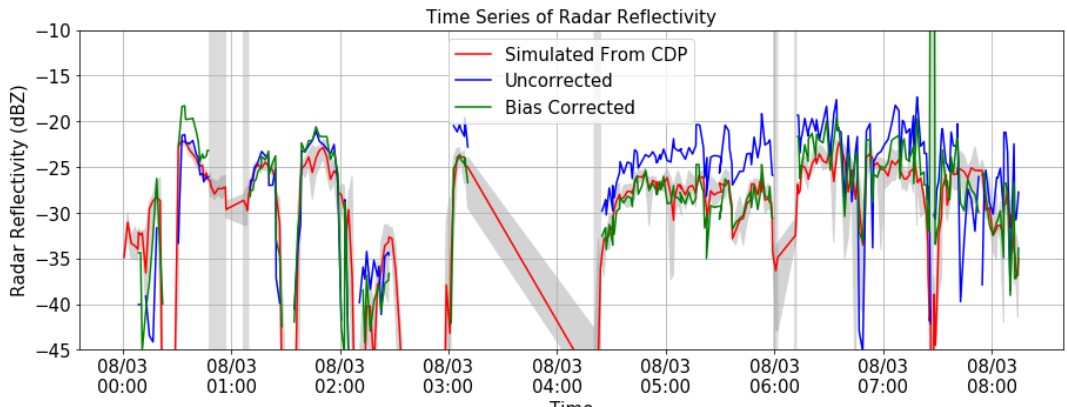

**Figure 5.** The radar reflectivity simulated from the CDP droplet measurements, observed with the BASTA cloud radar, and the observation after a bias correction had been performed.

Measurements from the CDP were averaged by total flow volume over each 10 second period. Although the binned distribution of liquid water droplet diameters is provided by the CDP, only the total number concentration and the
LWC was considered in this study.

One issue with using radar reflectivity to make retrievals of liquid water content can be the presence of pollen, insects and other non-meteorological airborne particles (sometimes referred to as airborne plankton). These can affect radar reflectivities in the boundary layer, and can be difficult to distinguish from signal caused by clouds. In this study, to ensure that radar reflectivity observations were caused by cloud droplets only, the cloud base height
and visibility measurements were used to ensure that only cloud radar reflectivities corresponding to either a fog or a cloud layer were used.

The minimum observable radar reflectivity is found from the signal to noise ratio. This therefore depends on the height of the gates, as both the noise and the maximum reflected power change with altitude. The sensitivity of the radar was found to range from $-52\,dBZ$ at $75\,m$ to $-33\,dBZ$ at $1000\,m$ (gates above $1000\,m$ were not investigated
due to the very small amount of radar gates not affected by ice particles above that altitude).

In order to investigate potential biases in the cloud radar reflectivity or radar simulator, figure 5 shows the comparison between the simulated reflectivity from the CDP measurements and the BASTA radar reflectivity. In this study, both the LWC and droplet number concentration from the CDP are directly used in the radar simulator. It can be noted that significant discrepancies can be observed between the two reflectivities, as observed between
4h30 and 6h UTC with a large under-estimation of the simulated reflectivity from the CDP measurements compared to the BASTA measurements.

Though the exact reason for this is not perfectly known, it could come partially from temperature dependencies of certain components of the radar (Toledo et al., 2020), from the CDP underestimating the LWC (through missing



**Figure 6.** Time series of a) radar reflectivity observed by the BASTA cloud radar in the background and equivalent radar reflectivity factor simulated from measurements of LWC and droplet number concentration made with the CDP in coloured circles, retrievals of a) temperature profiles and b) the LWP from the HATPRO microwave radiometer for the fog event observed on the 7-8th March 2020.





droplets or the mis-sizing of droplets for example) or other unaccounted-for effects. It should be noted that the max-
imum droplet size observable by the CDP is 50 µm. Due to the fact that the larger droplets have a disproportionately
large impact on radar reflectivity, the presence of these droplets could have contributed to this effect. In fact, Faber
et al. (2018) have attempted to characterise the uncertainty of the CDP instrument in studies using both glass beads
and water droplet generators, wherein a known size distribution is observed by the instrument and measurements
are compared to the distribution. Two common problems experienced by the CDP are coincidence error- where two
droplets cross the beam at the same time and are interpreted to be one droplet, and mis-sizing, normally caused by
droplets traversing the edge of the beam and interpreted as smaller droplets. The liquid water content observation
from the CDP is calculated from the sum of the mass of all droplets observed within the sampled volume, hence
the uncertainty in droplet diameter and concentration results in an even greater error for measurements of LWC, as
the LWC is proportional to the third power of droplet diameter. An additional phenomenon was also observed by
Russchenberg et al. (2004) and hypothesised to be due to cloud inhomogeneities at a scale smaller than the cloud
radar sampling volume. These could be particularly pertinent at the fog top height, where radar gates could be only
partially covered by LWC. In regards of the observed biases between the BASTA and CDP simulated reflectivities,
and in order to be able to validate 1D-Var retrievals with CDP measurements consistent with the cloud radar obser-
vations, a time-dependant bias correction was performed on the radar observations. To that end, the reflectivity bias
is computed as the 60 minutes average of the differences between the cloud radar and CDP simulated reflectivities
and then subtracted from the received power. Figure 5 shows the overall improved agreement between the CDP
simulated and the cloud radar reflectivities after bias corrections. The bias-corrected radar reflectivities are then
used in the next sections.

Brightness temperatures from MWR observations were also bias-corrected with a similar method as detailed in
Martinet et al. (2017, 2020). This method is based on a daily monitoring of differences between observed brightness
temperatures and simulated brightness temperatures from the AROME model 1 hour forecasts during clear-sky
conditions only. This MWR unit was also calibrated with the use of liquid nitrogen just before the experiment. The
obtained bias correction is thus relatively small with absolute values from 0.1 K to 1.8 K, the largest biases being
due to spectroscopic modelling errors in the oxygen band.

## 510   4.2   Case Study

In this study, one fog event taking place on the 7-8th March 2020 was examined due to the length of the event and
the thickness of the fog layer. This fog event had a large spread of LWC values, as well as going through phases of
thinning, a phase with a cloud aloft, and a phase of lifting from the surface before dissipation. The event also had
a relatively large duration of 9 hours, with in-situ measurements made over 11 hours.
In general, reliable radar signal is available from the third gate (37.5 m) during the night and the fourth gate 50 m
during the day. It was also shown in the synthetic data study that despite the liquid water path retrieval having a
smaller errors compared to the HATPRO microwave only retrieval, an error of $11.5 \, \mathrm{g \, m^{-2}}$ was present in retrievals

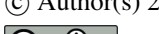



of the liquid water path. For this reason, the retrievals of liquid water content were expected to show good results of liquid water content only for fog layers with at least a thickness of 50 m.

As may be seen from the radar reflectivity observation shown in figure 6, an initial thin low cloud present around 21:00 UTC lowered to the surface to form a thin fog layer. This grew in thickness as cooling throughout the night took place. Observations with the tethered balloon began at around 23:00 UTC and continued until 10:00 UTC the following day, at which time the fog layer had lifted to form a low stratus cloud before completely dissipating. Figure 6 also shows the time series of retrieved temperature profiles and LWP from the MWR alone using the manufacturer

neural networks implemented in the instrument software. Pre-fog stable conditions with a thermal inversion close to the surface up to 3 UTC may be observed on this figure. Between 3 and 8 UTC, the temperature profile is almost iso-thermal within the first 300 m corresponding to the fog mature phase. Around 8 UTC, at the time of fog dissipation, the increase in surface temperature due to the heating of the sun may also be identified. The LWP retrievals show values below $20 \, \mathrm{g \, m^{-2}}$ until 0 UTC with then a large increase up to $80 \, \mathrm{g \, m^{-2}}$ during the fog mature

phase.

The radar and radiometer were situated in close proximity but were separated from the tethered balloon by a distance of up to 300 m. This distance varied as a function of the balloon height and the wind speed. From a scanning version of the cloud radar stationed close to the supersite, large variations in values of radar reflectivity were observed with horizontal spatial displacements. To ensure that comparisons of measurements made by the CDP

were therefore comparable to the retrievals, it had to be ensured that all instruments were observing a similar fog layer. In order to do this, a screening procedure has been applied to remove from the statistical analyses cases from which the BASTA cloud radar differ from the CDP simulated reflectivity by more than 3 dB. This difference took into account the instrumental and radar reflectivity simulator errors, meaning that differences of more than this must come from one of the previously mentioned effects which could account for differences between observed reflectivity

and that simulated from the CDP. The most likely explanation between the discrepancy between the observed radar reflectivity and the radar reflectivity simulated from the CDP was postulated to come from inhomegeneites, as highlighted in section 4.1.

In figure 7, the AROME forecasts corresponding here to the closest time (within a 3 hour windows) and location to the observation as well as the 1D-Var retrievals are shown. Also shown is the MRP background with the retrievals

using this as a background. With reference to the radar reflectivity shown in 6 a), it is evident that the presence of LWC is better modelled in the retrieval compared to the AROME background.

The initial lowering of cloud at 21:00 UTC is clearly seen in the retrieval, whilst the fog event begins with a thin layer at the surface in the model at 22:00 UTC. From the CDP measurements of the fog event investigated, a noticeable variation in the total droplet number concentration was observed, shown in figure 8. It can be seen

that the number concentration frequently diverges from the assumed concentration of $150 \, \mathrm{cm^{-3}}$ used in the radar simulator, from zero (evidently in the absence of LWC) to a maximum of $300 \, \mathrm{cm^{-3}}$ which was observed in the middle of the fog layer where the LWC was relatively large (approximately $0.3 \, \mathrm{g \, m^{-3}}$). The lowest values of number

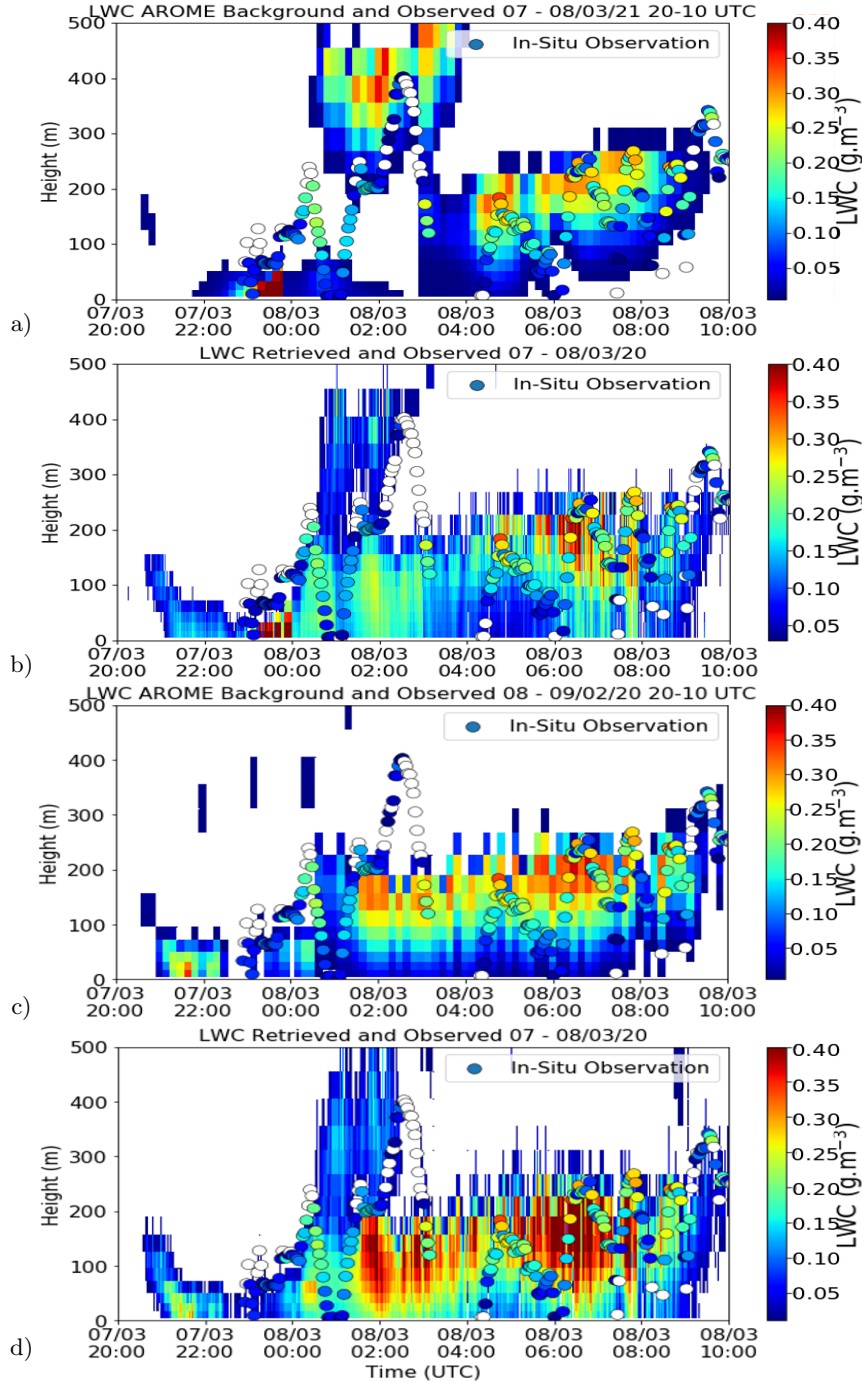

**Figure 7.** Background and Retrievals of LWC from the fog case on the 7-8th March 2020 using the nearest background (top row) and the MRP background (bottom row). Figure a) shows the LWC predicted by the AROME model; b) shows the retrieval of LWC made with the 1D-Var algorithm using both radar and microwave radiometer measurements from the nearest background; c) shows the MRP background and d) shows retrievals as made from the MRP background. The circles in all plots show the in-situ measurements made by the CDP of LWC.





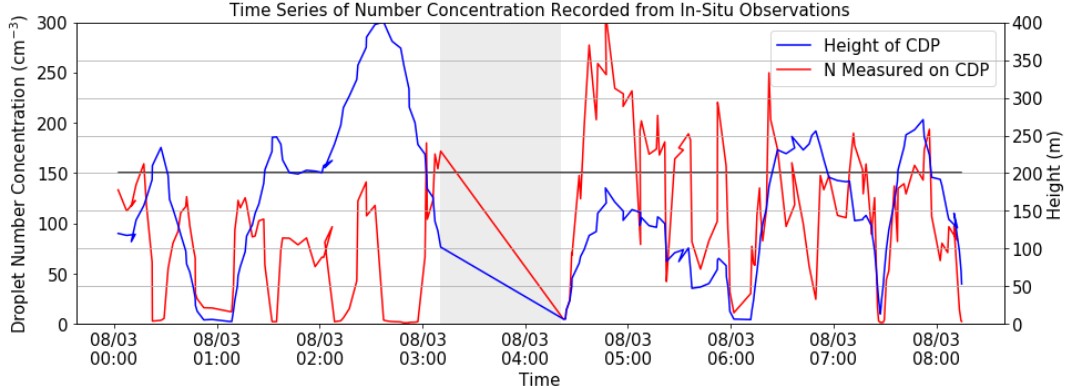

**Figure 8.** The total cloud droplet number concentration recorded by the CDP throughout the fog event observed on the 7 - 8 March 2020, with the height of the instrument shown on the same graph. The number concentration assumed in the retrieval of $150\,cm^{-3}$ is highlighted with the black line.

concentration, where the CDP was inside a cloud or a fog layer with a LWC above $0.5\,g\,m^{-3}$, were around $50\,g\,m^{-3}$- significantly higher than the lower bound of number concentration of fog events observed in some studies (Mazoyer et al., 2019).

### 4.3 Quantification of Improvements in Real Dataset

The retrievals were made with different configurations at a one minute time resolution, in order to estimate the sensitivity of the retrieval to certain parameters. In this section, 1D-Var retrievals will be directly compared to CDP LWC measurements when the BASTA reflectitiy and the CDP simulated reflectivity are comparable. Here, the two observations were considered comparable for a radar and CDP observation made at the same time and valid at the same height, with a difference in the simulated and directly observed radar reflectivity of $3\,dB$. However, several CDP measurements at ten second resolution are located within each 1D-Var gate with a variable grid size between around $10\,m$ at surface and $80\,m$ at $1\,km$. To take into account the CDP measurement variability within each 1D-Var gate, the 1D-Var LWC retrieval is compared to the median CDP measurements within certain spatial and temporal bounds. To that end, only CDP measurements within $10\,m$ of the height corresponding to the middle of the 1D-Var gate are taken into account in the calculation of the median. If the time needed by the tethered balloon to sense the whole +/- $10\,m$ bin is spread overall several minutes, an average of the 1D-Var retrievals within the time of ascent of the tethered balloon are directly compared to the median CDP LWC measurements.

The sensitivity of the 1D-Var algorithm to different settings is investigated by quantifying the impact on the LWC accuracy.





**Table 5.** The standard deviation and biases (observation - background/retrieval) of the retrievals using different background profiles.

|  | Nearest Profile | | MRP Profile | |
| --- | --- | --- | --- | --- |
|  | **STD** | **Bias** | **STD** | **Bias** |
| **Background** | 0.082 | 0.047 | 0.080 | -0.019 |
| **Retrieval** | 0.064 | -0.028 | 0.073 | -0.040 |

We first examine the effects of an improved background profile, found by using the MRP method to select the AROME background profile the closest to the observation within a 27 km domain and 6 hour time window (Bell et al., 2021). To this end, retrievals were made by with the MRP background and the background corresponding to the nearest time and location of observation. For the case in question, it is to be noted that the closest AROME model in time and location was able to forecast fairly well the overall structure of the second half of the fog event (figure 7), in which most of the in-situ observations were made. Because of this, the standard deviation of LWC errors in the nearest background profiles, found through comparisons to the in-situ observations, was very similar to that of the MRP background with values close to $0.08\,\mathrm{g\,m^{-3}}$ (table 5). As shown in table 5, the standard deviation of LWC errors was reduced for retrievals using both background profiles. The observed impact was, however sligthly better when the nearest background profile was used, with a $0.009\,\mathrm{g\,m^{-3}}$ lower standard deviation of errors and a lower bias by $0.012\,\mathrm{g\,m^{-3}}$. The differences in LWC retrieval statistics with both backgrounds are probably small enough to be considered within the spatial variability and uncertainty of the CDP measurements within each radar vertical gate, which was found to be $0.02\,\mathrm{g\,m^{-3}}$. As mentioned earlier, it is possibly due to the reasonably well-described fog structure from the nearest AROME background when the CDP measurements were performed during this unique fog event which prevents the benefit of the MRP method being seen. However, further analysis was performed concerning the better performance of the 1D-Var algorithm using the nearest profile compared to retrievals using the MRP background despite the larger number of clear-layers in the nearest background profile at range gates covered by fog or low clouds in the observation.

As was highlighted in section 2.2, one possible reason arises from the calculation of the Jacobian matrix which was handled so that diagonal components of the matrix could not be zero as it should be the case when the background profile is clear. Forcing Jacobians values to be non-zero even during clear-sky conditions was found to improve our results but probably hampers the benefit of the MRP which successfully corrected the fog vertical structure in the first part of the event. In fact, it can be noted that, at the beginning of the fog event, the nearest profile tended to underestimate the number of levels containing non zero LWC, whilst the MRP was able to better approximate the cloud structure as expected. The impact of not being able to force non zero Jacobians values where the initial hydrometeor contents of a pixel is zero was then investigated, as this is generally the case with variational frameworks using tangent linear approximation of the forward model due to the faster computation time. To that end the MRP





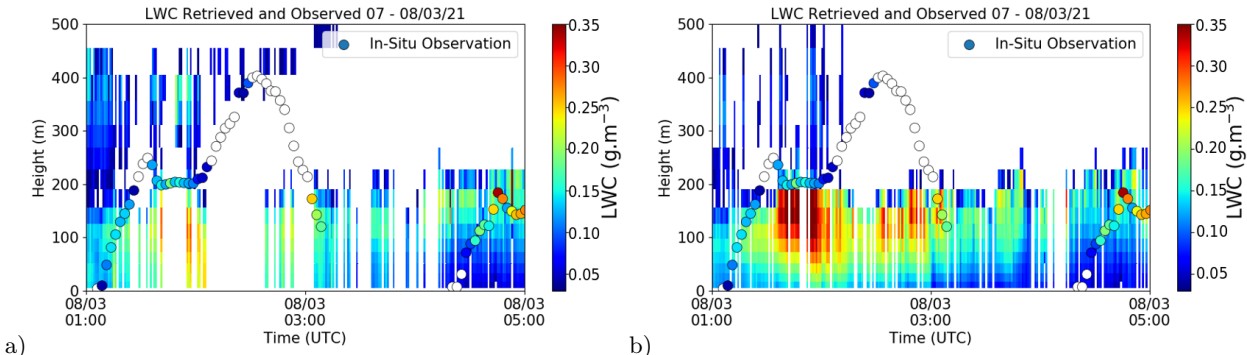

a)                                                    b)

**Figure 9.** Retrievals made with the a) nearest profile background and b) MRP background for the same fog event on the 7-8/03/2020, between 01:00 UTC and 05:00 UTC. The retrieval is shown in the background while in-situ observation are shown in circles. Where the retrieval failed to converge, the profile is left white.

**Table 6.** The convergence rate and median number of iterations for retrievals made with the MRP and the nearest background profiles for the fog case on the 7-8/20/2020. The **all non-zero** column refers to retrievals made with diagonal components of the Jacobian matrix made to be always non-zero, whilst the **zero below sens** column refers to retrievals made with the Jacobian matrix allowed to be zero where simulated radar reflectivity is below the sensitivity.

| | All Non-Zero | | Zero Below Sens | |
|---|---|---|---|---|
| | Convergence Rate | Median Iterations | Convergence Rate | Median Iterations |
| **Nearest** | 99.5 % | 8.0 | 84.0 % | 14.0 |
| **MRP** | 99.3 % | 4.0 | 87.8 % | 12.0 |

method was again compared to the use of the nearest profile when the Jacobian values were this time set to zero when the the background is clear.

Figure 9 highlights that for times when the nearest background poorly predicts the pixels containing LWC, the ability to replicate cloud in the retrieval is greatly degraded compared to when the Jacobians are forced to be non-zero (see figure 7 for comparison). Table 6 shows that whilst the algorithm is able to converge more that 99 % of the time for both background profile methods when the Jacobians are always non-zero, this is reduced to 84 % and 88 % when allowing Jacobians to be zero for the nearest and MRP backgrounds respectively. Another point to note from the table is that the median number of iterations needed for convergence was improved and even divided by two where non-zero Jacobians are forced through the vertical profile when using the MRP method, meaning that less time and computational power is needed to make the retrievals thanks to the MRP method. Further evaluation will be performed in the future to investigate deeper the benefit of the MRP method especially during cases when the closest AROME background in time and locations show larger discrepancies with the observed reflectivity.





Due to the better performance of the nearest profile on this specific fog case, however, the next evaluation of different 1D-Var configurations was performed using the nearest profile and allowing non-zero LWC Jacobians even when the background profile is clear.

Figure 10 shows the scatter plots comparing both the AROME background and the retrieved LWC of three configurations of the algorithm to the CDP LWC measurements. Comparisons were made by finding the median

LWC observed within $120\,\text{s}$ of the retrieval and within $10\,\text{m}$ of the retrieval height level. Error bars marked on the plots show the minimum and maximum values of LWC observed by the CDP over this time range and height difference for the retrieval in question. It may be seen that for certain comparisons, a wide range of LWC values are observed, which leads to uncertainties in the accuracy of the comparisons of $\pm\,0.15\,\text{g m}^{-3}$.

Whilst after 6:00 UTC, fog was predicted in the model, with values that were fairly close to those observed, the

fog event began with a large cluster of values where no LWC was predicted but was observed. Figure 10 b) shows the retrieval made using the nearest background profile improves all statistical measures of the LWC field with a decreased RMSE from $0.096\,\text{gm}^{-3}$ in the background to $0.066\,\text{gm}^{-3}$ in the 1D-Var retrievals. The correlation is also significantly increased from 0.56 to 0.72.

Two other configurations tested are also shown in figure 10 c) and d) to evaluate the impact of the background

error covariance matrix cross-correlations and the impact of microphysical assumption errors in the radar simulator. Figure 10 c) shows the retrievals made with a block diagonal $\mathbf{B}$ matrix in contrast with a full $\mathbf{B}$ matrix with cross-correlations adapted to fog events. Though an improved LWC field is obtained even with a block-diagonal background error covariance matrix compared to the AROME background, it can be noted that the 1D-Var analyses are slightly degraded compared to the retrieval with a fully correlated $\mathbf{B}$ matrix. In fact, the largest degradation is observed

on the correlation coefficient which is decreased from 0.7 to 0.6. This could suggest that the information from the increments in the temperature and humidity profiles can provide information useful to the retrieval of LWC as long as proper cross-correlations between variables can be defined.

Another configuration to be investigated evaluated the impact of forward model errors due to inaccurate approximation on droplet number concentration. To that end, the total droplet number concentration initially fixed at

$150\,\text{cm}^{-3}$ was changed to the observed number concentration from the CDP (figure 8).

As can be seen in figure 10 d), a significant improvement in the 1D-Var retrievals is observed. In fact, the RMSE is decreased from $0.066\,\text{g m}^{-3}$ to $0.049\,\text{g m}^{-3}$ and the correlation coefficient significantly increased from 0.7 to 0.8. This indicates that a non-negligible portion of the retrieval error is indeed due to the errors in the assumptions of size distribution of droplets. This sensitivity study demonstrated that optimal 1D-Var retrievals with increased

accuracy can be obtained when forward model errors can be limited and an optimal background error covariance matrix is used.

The impact of synergistic benefit of using the two instruments in real world conditions was also investigated. As is shown in figure 11, retrievals were run with the same configurations shown in figure 10 b), but with only one instrument. Figure 11 a) shows liquid water content retrievals made using only the microwave radiometer, and







**Figure 10.** Scatter plots of the liquid water content a) background and b) ,c) ,d) retrieved versus the liquid water content observed through with in-situ sensor. Panels b, c and d refer respectively to 1D-Var retrievals run with the nearest background profiles and full fog **B** matrix with cross-correlations between variables, 1D-Var retrievals run with the nearest background profile and a bloc-diagonal **B** matrix, 1D-Var retrievals run with the nearest background profiles and full fog **B** matrix with cross-correlations with the use of the CDP droplet number concentration inside the radar simulator during the minimization. Error bars indicate the range of in-situ observations recorded throughout the height level corresponding to the retrieval height levels.



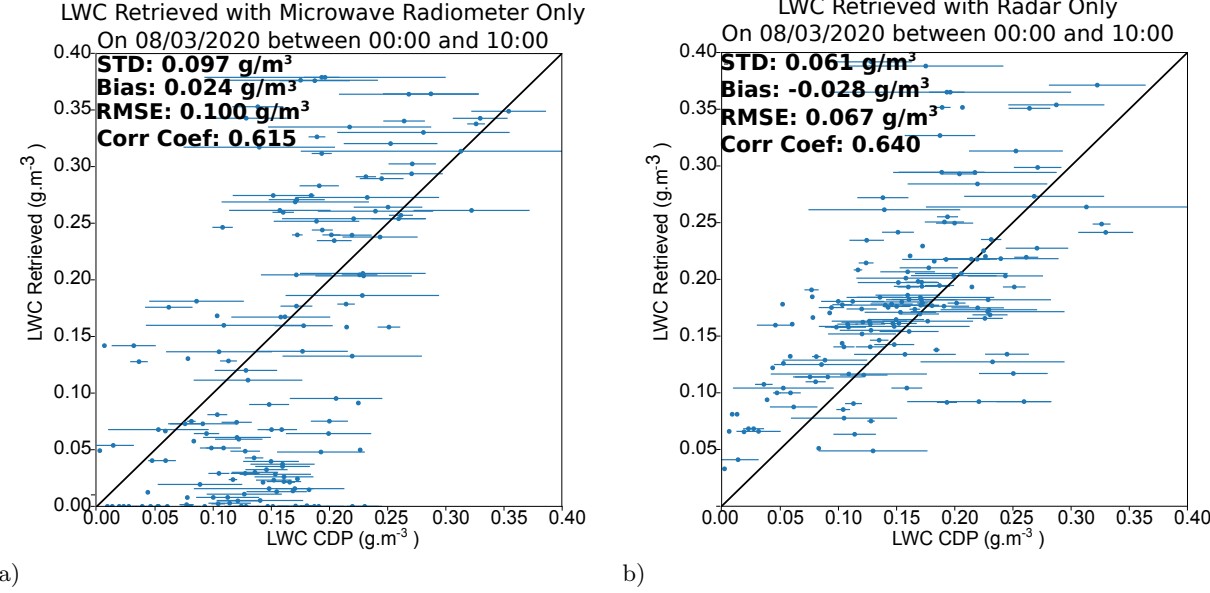

**Figure 11.** Scatter plots of the liquid water content retrieved versus the liquid water content observed through with in-situ sensor. Panels a) and b) refer respectively to 1D-Var retrievals run with the only the microwave radiometer and only the cloud radar, both with the nearest background profiles and full fog **B** matrix with cross-correlations between variables. Error bars indicate the range of in-situ observations recorded throughout the height level corresponding to the retrieval height levels.

figure 11 b) with only the cloud radar. As expected, the radiometer is not able to well describe the distribution of liquid water content alone, and only small increases in the correlation coefficient, from 0.57 to 0.62, are found when compared to the AROME model. With the radar only configuration, results were slightly degraded compared to those found from the synergistic retrievals, with a correlation coefficient of 0.64 compared to 0.71 and a standard deviation of errors of 0.61 compared to 0.60 in the synergistic setup. One-instrument retrievals of temperature and humidity

at the times of the radiosonde launches were also performed to investigate whether any benefit could be added to the radiometer retrievals through the inclusion of a cloud radar. As was found in the synthetic dataset studies, the retrievals made with only the radiometer were not statistically worse than with both instruments, indicating that information from the cloud radar does not significantly contribute to retrievals of humidity and temperature.

It may of course be seen from the errors shown from the fog case in question that the retrieval errors were larger

than those found from the synthetic dataset study. This was expected and there are several factors that could contribute to this. The first is that in the synthetic study, it is assumed that the background and observation errors may be perfectly modelled. The dataset of background profiles will contain the errors specified in the **B** matrix, and similarly the observational dataset will contain the errors specified in the **R** matrix. However, these matrices are only estimations of the true errors. Additionally, observations are supposed to be un-biased which might not be the





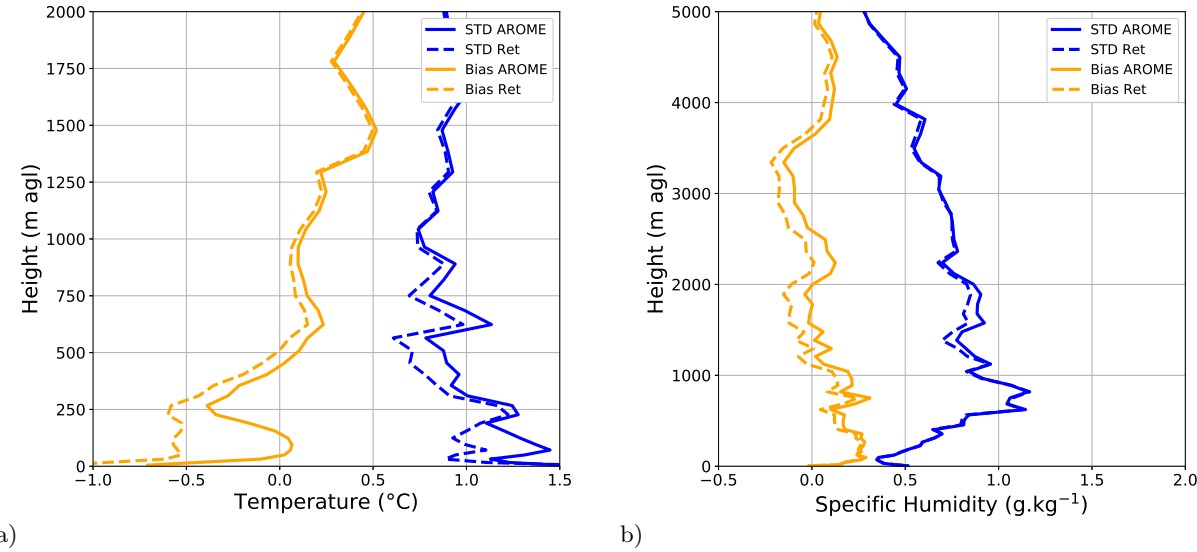

a)
b)

**Figure 12.** RMSE and bias of a) temperature retrievals and b)specific humidity retrievals made throughout the SOFOG-3D field campaign. Dotted lines correspond to the background profile and dotted lines to retrieved profile statistics, where blue lines denote the standard deviation of errors and yellow lines to the bias.

case- even though a bias correction is proposed in this study, it may not be optimal. Another reason for the apparent increase in errors in this study compared to the synthetic data study probably comes from the fact that the in-situ measurements used in this study themselves contained substantial error. Indeed, the error in LWC measurement by similar CDP instruments has been estimated to be up to 50 % (Wendisch et al., 1996) (equivalent to up to a 3 dB radar reflectivity error when using the modified gamma distribution) . The variability in the measurement can

be seen from the error bars on the scatter plots, with the range of CDP LWC measurements in some cases being greater than 100 % of the median observed value. An error in the comparison could also arise from the fact that the observations made from the balloon platform were of a much smaller sampling size, and in a location up to 300 m away from the radar and radiometer field of view. Though an attempt to reduce the impact of this error was made through the permitting of comparative statistics only when the difference between observed radar reflectivity and

radar reflectivity simulated from the CDP was less than 3 dB, a difference in size distribution in the droplets could not be ruled out.

Finally, to check the validity of temperature and specific humidity retrievals as expected from previous studies using MWR alone (Martinet et al., 2020), a verification was performed through checks against radiosonde observations. In the field campaign, radiosondes were launched up to four times per day, meaning that fewer points of comparison

could be made between observations and retrievals per fog event, compared to for the LWC observations from the





tethered balloon. To make the verification over a statistically significant number of observations, all radiosoundings launched during the campaign IOPs were thus taken into account.

It can be seen in figure 12 that in both temperature and humidity retrievals, the standard deviation of errors is reduced when compared to the nearest background profile, which was the AROME model valid at the time and location of the observations with a much higher impact for temperature compared to specific humidity . It may be noted, however, that both the temperature and the humidity retrievals do not improve by the magnitude shown in the synthetic dataset study. Whilst the temperature retrievals were more accurate than the AROME model throughout the boundary layer, this improvement tended to range from 10 % to 20 %, in contrast to the 40 % seen in the synthetic dataset study. For the specific humidity retrievals, the retrievals showed smaller improvements of up to 10 % between 1000 m agl and 2000 m agl.

As mentioned in Martinet et al. (2020), 1D-Var retrievals are quite sensitive to the choice of the background error covariance matrix especially for humidity retrievals and temperature retrievals within the first kilometer. One possible reason for the smaller improvement obtained on the temperature and humidity retrievals could thus come from a non optimal **B** matrix used during the minimisation. In fact, it should be noted that for the application to real observations, the same static fog **B** matrix valid for November 2018 has been re-used. However, our validation takes into account all the SOFOG3D radiosoundings launched at the super-site during atmospheric conditions prone to fog occurrences according to the AROME model forecasts but only 9 of them were found to be launched within a fog layer. The temperature and humidity retrievals might thus be representative of stratus clouds and clear conditions more than fog conditions making the fog SIRTA **B** matrix potentially sub-optimal. Further evaluation will thus be conducted in the future to adapt the fog **B** matrix to the time and location of the observation to be more consistent with the real observation errors.

## 5 Conclusions and Future Prospects

In this article, a methodology for the initial step towards the assimilation of cloud radar and microwave radiometer combined observations is presented. A description was given of the 1D-Var algorithm which is used to perform retrievals on temperature, humidity and LWC in vertical profiles.

A validation of previous work to improve the accuracy of the background profile was also attempted in this study. It was found that for the fog case investigated that the accuracy of LWC retrievals were not improved by using this improved background profile, called the MRP profile. One factor influencing this finding was the decision to force the Jacobian matrix to have non-zero diagonal values where they would otherwise have been calculated to be zero (especially in cases where the initial background is clear). When this configuration was not used, the MRP profile was seen to improve retrieval convergence and to better resolve the cloud structure compared to the nearest background profile, especially when the nearest background profile is not able to predict a fog or cloud layer when it is observed.





Additionally, in the fog case investigated, the AROME model nearest background tended to predict fog or low cloud for most times when these phenomena were observed at the supersite at the corresponding altitude of the CDP measurements. As the main aim of the MRP method was to correct the background errors for times at which a fog event is undetected due to spatial and temporal forecasting errors, it is possible that a larger benefit of the MRP method could be concluded with another fog case study where the nearest AROME background profiles shows larger errors.

With this in mind, it is recommended that other fog cases should be examined in the future to provide a more robust evaluation of the benefit of the MRP method to improve the 1D-Var retrievals.

It was shown that when retrievals were made using the assumed droplet number concentration of $150\,\mathrm{cm}^{-3}$, an overestimation of concentration was made for much of the fog event analysed. An overestimation of droplet concentration will lead to an underestimation in radar reflectivity with the radar simulator used inside the 1D-Var. This can lead to the algorithm incorrectly increasing, or not reducing by enough, the LWC in state vector $\mathbf{x}$. By changing the droplet number concentration in the radar simulator to that recorded by in-situ measurements, both the standard deviation and bias of retrieval errors were reduced by $0.010\,\mathrm{g\,m}^{-3}$ and $0.024\,\mathrm{g\,m}^{-3}$. This suggests one area in which the algorithm could be improved. Though there is little consensus on how cloud microphysical properties vary with fog properties, such as the height of the fog top, the change in distribution within the fog layer (at the top, middle or bottom), the LWP, turbulent properties or the stage in the fog life cycle (formation, mature, dissipation), more research into this topic could allow a parameterisation of the total number concentration, which if more accurate than the assumed number used in this study, could improve retrievals.

The inclusion of a new microphysical scheme could also improve the accuracy of the droplet size distribution specified in the retrieval. The Liquid Ice Multiple Aerosol (LIMA) scheme is a quasi two-moment microphysical scheme which adapts the droplet concentration number to the cloud condensation nuclei. Work is currently underway for this to be integrated into the operational model AROME. If this is able to provide better estimates of the total droplet concentration number than is currently assumed, this information could also be specified to the algorithm to improve the accuracy of retrievals.

It was demonstrated that the combination of cloud radar and microwave radiometer observations showed potential to give more accurate retrievals of LWC compared to the use of cloud radar observations alone. In the synthetic dataset study, the retrieval error was seen to decrease for synergistic retrievals compared to when only the radar was used, and the DFS for LWC was also seen to increase.

As the persistence and evolution of a fog event has been demonstrated to depend on the LWC of the fog event (Toledo et al., 2021), it is likely that the improved LWC field in a high resolution model can improve forecasts of the dissipation of fog events. The forecasting of stratus lowering fog events may also be improved through the improved representation of the presence and LWC distribution of low clouds, something which this framework could also contribute towards.

*Author contributions.* AB and PM made developments to the 1D-Var algorithm.AB performed the analysis documented in the paper which was supervised by PM and OC. JD and SJ provided the radar data, on which they performed verification checks and gave relevant assistance. FB provided the in-situ data from the SOFOG-3D field campaign. YS provided the configuration to generate AROME forecasts. VU provided treatment for the microwave radiometer data.

*Acknowledgements.* The instrumental data used in this study are part of the SOFOG3D experiment. The SOFOG3D field campaign was supported by METEO-FRANCE and ANR through grant AAPG 2018-CE01-0004. Data are managed by the French national center for Atmospheric data and services AERIS. The MWR network deployment was carried out thanks to support by IfU GmbH, the Köln University, the Met-Office, Laboratoire d'Aérologie, Meteoswiss, ONERA, and Radiometer Physics GmbH. MWR data have been made available, quality controlled and processed in the frame of CPEX-LAB (Cloud and Precipitation Exploration LABoratory, www.cpex-lab.de), a competence center within the Geoverbund ABC/J by acting support of Ulrich Löhnert, Rainer Haseneder-Lind and Arthur Kremer from the University of Cologne. This collaboration is driven by the European COST actions ES1303 TOPROF and CA18235 PROBE. Thibaut Montmerle and Yann Michel are thanked for their support on the use of the AROME EDA to compute background error covariance matrices.



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
