# Peer review of "An Optimal Estimation Algorithm for the Retrieval of Fog and Low Cloud Thermodynamic and Micro-physical Properties"

_Atmospheric Measurement Techniques, 2022_

## Author Comment (AC1)

**Article Reviewer Comments and Responses**

Thanks to both reviewers for taking the time to read the article, and asking some interesting and relevant questions on the research presented. The original comments and questions are written below in black, with our responses in green and the changes made to the paper marked in red.

Line 278: I thought it was interesting that you assumed a diagonal observation covariance matrix, especially since the Cimini et al. 2018 study showed important off-diagonal values in the MWR forward model (esp between 51, 52, and 53 GHz channels). Would adding off diagonal elements change the results much at all?

This is an interesting point, and one that has been tackled in different ways in the literature. The uncertainty affecting different spectroscopic parameters is likely to be related, and the modeling of different channel response to a set of atmospheric parameters is also likely to be correlated to a certain extent.

It is also possible that correlations will exist between the different radiometric channels as due to instrumental properties, as a result of calibration or channel response, for example. These can be more difficult to prescribe as they may change with time.

In this work, a diagonal observation error covariance matrix was assumed, which for radiometric measurements is in line with the works of Martinet et al. (2015;2017) and Ebell et al. (2017). The impact of non-diagonal terms in the observation error covariance matrix has been investigated by Pauline Martinet in the context of MWR retrievals without cloud radar data and no impact was found on temperature and humidity retrievals. This is probably related to the fact that the off-diagonal values are more affecting the channels with the largest observation errors due to larger errors in the radiative transfer model (and thus with less impact on the 1D-Var retrievals). However some studies have made use of a non-diagonal observation error covariance matrix for cloud radar observations such as Löhnert et al. (2004). A sensitivity study was conducted with the synthetic dataframe to examine the effect on the retrievals of a non-diagonal observation errors from Toledo et al. (2020). The standard deviation of differences in the retrieved profiles compared to the retrievals for a diagonal observation error matrix ranged from 0.020 to 0.015 g.m-3 between 50 and 200m. As this was seen to be small compared to the effects of changing the B matrix, this was assumed to be of less importance for the retrieval accuracy.

Line 354: I think adding a reference that demonstrates that the MWR is only sensitive to LWP (not LWC) here would be good. You might add: Crewell, S., K. Ebell, U. Loehnert, and D.D. Turner, 2009:Can liquid water profiles be retrieved from passive microwave zenith observations? Geophys. Res. Lett., 36, L06803, doi:10.1029/2008GL036934.

**Added as suggested.**

Figure 3e: why is the radar-only bias smaller than the MWR-only bias? And why is the synergistic bias the worst of all? Given this is a synthetic study, this can and should be understood.

Thanks for pointing this out. Indeed, this did appear somewhat strange and was investigated. As it was a synthetic dataset, in which the perturbations made to the real profiles to make the a-priori

followed the errors specified in the background error covariance matrix used to make the retrievals, close to zero bias was expected in the results. However, many times a perturbation would be made to the liquid water content field which made it less than negative. As this is un-physical, the liquid water content would be set to zero, thus increasing the average amount of liquid water content. This effect is seen in figure 5a) - a positive LWC throughout the profile, but largest in the bottom layers where the background errors for LWC are the largest (as a background error covariance matrices for fog cases was used).

In the retrievals, a full background error covariance matrix was used, with positive correlations between liquid water content and specific humidity. For many cases where the background LWC had been 'artificially' increased to zero in the background profile, the algorithm will not want to increase the humidity due to the positive correlations. As this effect does not happen when positive perturbations are initially made to the 'true' LWC when creating the background profiles, the net effect of this is a negative bias.

The reason why this bias is seen most with synergistic observations was due to the fact that when no MWR observations were given, there would be very little change to the humidity profile, hence no net bias. When only MWR observations were used, the contribution to the cost function of increasing the LWC at a given altitude was lower, as this was not in disagreement with another observation, and both LWC and specific humidity could be more easily increased. However, when radar observations indicated a correct LWC at a given altitude, but MWR observations indicated that an increase in humidity was required, this effect would be most strongly seen.

Figure 3f: Accidentally repeated the humidity bias figure, when you meant to show the humidity STD panel.

**Corrected.**

Line 411: You state "The measurement error will also mean that the information content from the observations will also decrease" – this is an odd phrasing. I think it would be better to say something like "The information content from the observations depends upon the measurement uncertainty in the observations, with larger uncertainties resulting in smaller amounts of information content."

Changed to: The measurement uncertainty will also affect the information content in the retrievals, with a larger uncertainty resulting in a lower information content.

Line 429: This finding (i.e., that the DFS from a cloud radar is about 35% for LWC) agrees very well with an earlier study by Ebell, and should be referenced: Ebell, K., U. Loehnert, S. Crewell, and D.D. Turner, 2010: On characterizing the error in a remotely sensed liquid water content profile. Atmos. Res., 98, 57-68, doi:10.1016/j.atmosres.2010.06.002.

Added: These results agree well with a previous study by Ebell et al. (2010).

Line 431: The total DFS for temperature and humidity depends on the vertical layer over which the DFS was computed. Please add the height range (e.g., surface up to 2 km) in this sentence please.

Changed to: When only MWR observations are used, with respective values of 2.31 and 0.75, the DFS between the surface and 30 km asl for temperature and humidity was smaller than that found in other works (Löhnert and Maier, 2012).

Table 3: why is the DFS for temperature from dual retrieval (1.99) less than for the MWR-only (2.31)? This does not make sense to me.

This was also investigated as it seemed not to make sense. It was clear from the retrieval errors in the synthetic study that the cloud radar lacked any meaningful sensitivity to either temperature or humidity. However, a reduction in DFS implies that it reduced the amount of information that could be used in retrievals when this was included in the retrieval method, which was not expected.

The DFS for humidity, did, however increase when the cloud radar was included in the retrieval routine. It was concluded that the cross covariances in the B matrix therefore had an impact on the total DFS found for each variable. Indeed as this was a synthetic study, the background profiles should have had perturbations to variables which would be correlated as described by the background error covariance matrix, however, going back to the point raised earlier, this was not the case. When perturbations to the liquid water content field decreased below zero, the field was set to zero, meaning that the background error correlations between liquid water, humidity and temperature were no longer those described by the B matrix.

It is also worth mentioning that the DFS here is the average DFS of all the retrievals made in the synthetic dataset. For many individual profiles, and for some profiles, the DFS for temperature and humidity was seen to stay constant or increase with the inclusion of the cloud radar.

Line 446: You indicate that most of the information for water vapor is in the 1 to 3 km range. I think it would be useful to indicate that this is well above the top of fog layers, and thus that the MWR really offers only limited water vapor information within a low-lying fog layer.

Added: It should here be noted that most humidity information brought by the measurements is significantly above the fog layer. For the SOFOG-3D field campaign, 80% of events measured with the tethered balloon had a maximum fog top height lower than 200m, thus a limited improvement could be expected.

Line 630: I totally agree with the commend on how proper cross-correlations between variables in the background will improve the retrieval. I think this is a great opportunity to also include the need to have improved measurements of the layer-to-layer covariance in fog (and cloud) properties directly from observations.

Yes this is a good point. If these retrievals were targeted towards only research into fog processes for which the proper correlations could be implemented in the right conditions (potentially different correlations for different types of fog, different air masses etc.) with a B matrix derived from observations for these conditions, this would be very relevant. Indeed, optimal estimation techniques often derive background error covariance matrices from observations (such as Cimini et al. (2010) ), which could very likely be improved through improved measurements. However, this methodology requires that long-term in situ measurements are made for each site, which constitutes a serious drawback. Indeed, we aim to use an ensemble data assimilation, in which background error covariances would vary according to the model state, and so it may be difficult to incorporate the observational findings into improvements in fog background error covariance matrices. In evaluating the efficacy of the ensemble data assimilation of generating representative background error matrices, however observations of a good quality will be vital.

**Added:**

Optimal estimation techniques often derive background error covariance matrices from observations (such as Cimini et al. (2010)), which could very likely be improved through improved measurements of fog and cloud properties. However, this methodology requires that long-term in situ measurements are made for each site, which constitutes a serious drawback. As explained in section 2.5, the **B** matrix for this work was produced through an ensemble data assimilation. It is thus likely that an improvement in the understanding and the modelling of fog processes in high resolution models such as AROME could lead to more accurate background profiles with smaller background error covariances between different variables and model layers in the B matrix. In the future, improved measurements of fog and cloud properties could be useful for evaluating the efficacy of the ensemble data assimilation of generating representative background error matrices.

Fig 12: please add that the bias and RMSE are computed relative to radiosondes in the caption

Added: RMSE and bias of a) temperature retrievals and b) specific humidity retrievals relative to radiosonde observations

Line 683: the temperature retrievals are more accurate in terms of a smaller RMS, but the bias is worse. I think that is worth mentioning here.

Added: It should also be noted that there was a degradation in the temperature bias below 500 m. My only major comment is that there are places in the paper where the uncertainty analysis could be improved. Some particular questions that arose when I was reading the manuscript: how much uncertainty is induced due to the variability of drop-size distributions given that the simulator assumes a modified gamma distribution?

Some analysis into the uncertainty of assuming a modified gamma distribution with a set of parameters prescribed for the ICE-3 microphysical regime was conducted in a previous paper (Bell et. al, 2021). In this paper, the errors in simulating radar reflectivity due to the errors in knowing the correct parameters of the droplet distribution were assessed to be 3.9 dB and 2.2 dB for a LWC of 0.12 gm-3. However, these errors increase for a larger LWC and are reduced for smaller quantities of LWC. This error is taken into account in the R matrix.

According to Rodgers (2000), the analysis error covariance matrix from model parameters may be found from (with the notation used in this paper)  $G_yH_bR_bH_b{}^TG_y{}^T$  where  $G_y$  is the gain matrix,  $R_b$  is the uncertainty in the model parameters, and  $H_b$  is the sensitivity of the forward model to the model parameters ( $\partial F/\partial b$ ). As the retrieval algorithm was not designed to calculate  $H_b$  (which would depend on the atmospheric state), this calculation was made by assuming that  $H_bR_bH_b{}^T$  was a diagonal matrix equal to the values found in the previous study, of 3dB for the cloud radar.

The results showed that the contribution of errors in the radar simulator model parameters contributed around 0.010 gm-3 to 0.025 gm-3 for a fog retrieval profile with a relatively high maximum LWC.

Fig 1: LWC from a retrieved profile with the estimated uncertainty resulting from the forward model parameters.

Added:

The retrieval uncertainty due to the forward model parameter uncertainty may also be investigated. As mentioned in section 2.3, several parameters must be specified to the radar simulator to prescribe the cloud droplet size distribution. As these parameters may not be representative of the observed cloud droplets, this induces a certain amount of error into the retrieval. From Rogers (2000), the analysis covariance of the errors in the retrieval due to error in the forward model parameters, **A**b, may be calculated from equation 6.

$$\mathbf{A}_b = \mathbf{G}\mathbf{H}_b\mathbf{R}_b\mathbf{H}_b^T\mathbf{G}^T \tag{6}$$

$$\mathbf{G} = (\mathbf{H}^T \mathbf{R}^{-1} \mathbf{H} + \mathbf{B}^{-1})^{-1} \mathbf{H}^T \mathbf{R}^{-1}$$
(7)

where  $\mathbf{R}_{\mathbf{b}}$  is the covariance matrix of forward model parameter errors,  $\mathbf{H}_{\mathbf{b}}$  is the sensitivity of the forward model to the prescribed parameters (analogous to equation 4) and  $\mathbf{G}$  is called the gain (or contribution function) matrix.  $\mathbf{A}_{\mathbf{b}}$  represents the error covariances in the retrieval due to the assumptions about the forward model parameters. In Bell et al. (2021), an analysis of expected error in two of the droplet size distribution parameters- the total droplet concentration N and the shape parameter v- was conducted with the aid of previous literature on fog and cloud droplet distributions. These uncertainties can be used to create the matrix  $\mathbf{R}_{\mathbf{b}}$ . In this study, it was found that the total droplet concentration could be expected to range from 30 to 300 cm-3 and the shape parameter from 2.5 to 15. It was assumed that this matrix was diagonal i.e. that the error in the two parameters was not correlated, and was not correlated between different retrieval height levels.

In order to avoid directly calculating  $H_b$ , the reflectivity from a profile can be simulated with the radar simulator, and then once again by making a small perturbation to the simulator parameters. By finding the difference between the first and second simulations, and dividing this by the perturbation size, the matrix  $H_b$  may be approximated – in a manner as was explained in section 2.2.

The retrieval error resulting from the errors in the forward model parameters was estimated for one profile with a maximum LWC of  $0.3 \text{ g.m}^{-3}$ . From the square root of diagonal components of matrix  $A_b$ , calculated from equation 6, model parameters were found to contribute between 0.01 and 0.025 g.m-3 to the total retrieval error for liquid water content. If the droplet concentration would be known, the contribution of forward model parameter error fell to between 0.006 and 0.014 g m-3.

Were any drops larger than the 50 um maximum size detected by the CDP present (how would this be known) and could this have any impact on the comparison with the in-situ measurements?

There is a chance that droplets with a maximum size higher than 50 µm were observed by the CDP during the stratus phase between 5 and 8 UTC. We can make this assumption due to the underestimation of simulated reflectivity from the CDP measurements compared to the BASTA reflectivity that might comes from the presence of large droplets having a large impact on the reflectivity. However, this phenomenon has also been documented in Russchenberg (2004) and has been proposed to come from the spatial representation of the cloud radar compared to the in-situ measurements (the sampling volume of a cloud radar is much larger than the in-situ sensor), and the assumption of microphysical homogeneity inside the cloud radar sampling volume may lead to biases when comparing the two.

To estimate the presence of droplets above 50  $\mu$ m, the CDP size distribution could be checked, and if the CDP measures droplets of ~40/50  $\mu$ m, there would be a high chance that the size distribution has been truncated due to its detection limit. When this was done, it was found that for the largest liquid water contents observed with the CDP instrument, the mode of the distribution ranged from around 10  $\mu$ m to 20  $\mu$ m, with fewer observations of droplets larger than 30  $\mu$ m. However, when a larger radar reflectivity bias was present between the radar observations and the reflectivity simulated from the CDP measurements, the number of droplets detected above 30  $\mu$ m did increase.

In fact, as shown in figure 3, showing the CDP measurements made at 7 UTC that day, the modal droplet size ranged from around ~10  $\mu$ m – ~22  $\mu$ m. At the top end of this range, the modal droplet diameter was higher than that predicted by the droplet distribution assumed in the radar simulator, and a larger number of droplets are likely to have been missed than predicted by our assumed droplet distribution.

**Could this have any impact on the comparison with the in-situ measurements?**

For the droplet distribution assumed in the radar simulator with an LWC of .12gm-3, drops larger than 50 µm account for less than 0.15% of the total LWC. Figure 2 shows the contribution towards the total LWC of droplets of different sizes for a total LWC of .12gm-3. A more likely issue for the comparison between retrieved LWC and in-situ measurements would be the presence of drizzle. Drizzle drops are often seen where the cloud droplet diameters are large, and they would contribute disproportionately highly to radar reflectivity, but not be measured by the CDP. For times at which larger droplets were present than predicted by the assumed size distribution, a truncated spectra may have been measured by the CDP which could have contributed to (perceived) LWC retrieval error.

Fig 2: Percentage of Total liquid mass contributed by droplets of that Diameter vs Droplet Diameter for a LWC of 0.12 gm-3 from the modified gamma distribution used in the radar simulator inside the retrieval algorithm.

Fig 3: Costabloz (personal communication, June 2022). Microphysical observations from the CDP mounted on the tethered balloon for an ascent between 06:55 and 7:08 UTC on 08/03/2020.

Added: Although the CDP can only measure droplets of up to 50  $\mu$ m, few observations were recorded of cloud droplets with a diameter over 40  $\mu$ m, and the assumed droplet size distribution predicted that droplets over this size would account for only 0.15 % of the volume of cloud liquid water observed, for an LWC of 0.12 g.m-3. However, the modal diameter sizes observed often had a non-negligible difference compared to that predicted by the assumed droplet distribution in the forward model (for the same LWC). It is therefore possible that a larger proportion of droplets with a diameter bigger than 50  $\mu$ m were present and not observed by the CDP.

Can you be more quantitative in the uncertainties associated with the bulk parameters derived from the CDP probe?

The quantification of error in the CDP (or FSSP, a prior, similar instrument (Baumgardner and Spowart, 1990) has been attempted through numerous methods including laboratory verifications with glass beads and droplet generation systems (Baumgardner et al. 2017) and comparisons to instruments working on different measurement principles, such as hot wire probes (Wendish, 1996; Baumgardner et al. 2017). Results showed that the uncertainties will depend on the particle size but that sizing uncertainties are in general under 15%. When calculating the LWC from these measurements, the uncertainty is larger, however, as this is calculated from the third power of droplet diameter and the concentration of particles, which also contains an uncertainty.

Furthermore, most uncertainty experiments have been done from an airplane platform. The CDP used in this experiment was modified to contain an aspirator to allow an air flow for droplets. The flow speed is vital for the calculation of droplet concentration. A thorough analysis of the expected errors from CDP observations from a tethered balloon platform is being conducted by F. Burnet and M. Fathalli, and will be published soon. From personal communication with them, the following errors may be assumed:

~ 25% for N (droplet number concentration)
~ 30 % for LWC

Are these instrumental/measurement errors larger or smaller than the variability that is used to quantify the errors in Figure 10?

Looking at the error bars of figure 10, which are not representing the CDP instrumental errors but only the variability of CDP measurements within each radar vertical bin, it may be seen that in general there is a variability of +/- 0.1 g/m3 within the radar bin. Depending on the median values of the LWC, it means that the measurement variability is between 100% for the smallest values of LWC (of around 0.1 g/m3) and 30% for the highest values of LWC (around 0.3 g/m3).

From the above estimation of CDP LWC observation error, of around 30%, the recorded LWC variability, used to quantify the errors in figure 10, is roughly in line with the expected LWC uncertainty. The CDP observation errors are, however, smaller for the lowest values of LWC.

See: Indeed, the error in LWC measurement by similar CDP instruments has been estimated to be up to 50% (Wendisch et al., 1996). Compared to the CDP measurement variability which was used in Figure 10 to quantify the error when comparing the retrieved LWC with the CDP LWC, the expected CDP measurement errors are of the same order of magnitude as the CDP variability for the highest values of LWC but they should be smaller for the lowest LWC values.

**Can you give an error in the retrieved LWC based on whatever uncertainties in assumed parameters that go into the retrieval algorithm?**

This was something that was investigated in a little more detail in the previous paper (Bell et al., 2021). In this paper, a summary of other literature examining the expected errors for parameters inside fog and stratus clouds is presented. Values of number concentration are expected to range from 30-300 for fog and the shape parameter is expected to range from 2.5 to 15. Radar reflectivity was simulated for the 25th to 75th percentile of these values and the assumed error taken from the range in these values. This resulted in a forward operator parameter error of approximately 3 dB when the 25th to 75th percentiles of these values were considered. The results showed that the

contribution of errors in the radar simulator model parameters contributed around 0.010 gm-3 to 0.025 gm-3 for a fog retrieval profile with a relatively high maximum LWC. However, this was done for a typical profile. The method is therefore not adapted to different LWC values.

From the same profile for which uncertainties were given before, when all uncertainties are considered (instrumental + forward operator) the maximum uncertainty increases to 0.032 g m-3.

Is there any concern about the attenuation of the radar signal? I know the retrievals are not attempted when rain is present because the attenuation will be greater, but can any estimate be made on the attenuation effects due to the liquid water content itself?

Indeed, the problem with rain presence is not only the attenuation, but the fact that the signal becomes dominated by rain droplets, and it is not possible to distinguish between the reflectivity caused by the rain droplets from those caused by cloud droplets. Rain also increases the attenuation due to droplets on the radome, which is equally difficult to model. The attenuation on the radome can be estimated with reference to a fixed metal calibration target, which was done for the SOFOG campaign. However, at the time of writing, this has be seen to not sufficiently estimate the attenuation.

Alternative methods also exist to estimate the attenuation from a water layer on the radome from changes in the radar noise level (Fabry, 2001).

The attenuation from liquid cloud is taken into account in this algorithm. Values tended to agree with the results found by Tridon et al. (2020) who found attenuation values for 94 GHz cloud radar of 4 dB km-1/gm-3 for cloud droplets in the Rayleigh regime at a temperature between 0°C and 10°C.

Could a phrase or sentence be added to the abstract describing how the synthetic dataset is constructed?

Added: This dataset was constructed by perturbing a high resolution forecast dataset of fog and low cloud cases by its expected errors.

It is nice that the abstract gives the quantitative uncertainty in LWC. It would be nice also to state the fractional uncertainty to give a better idea on the size of the error bar.

Added: With real data, as expected, retrievals with a good correlation (0.7) to in-situ measurements, but with a higher uncertainty than the synthetic dataset, of around 0.06 g.m-3 (41 %), was found. This was reduced to 0.05 g.m-3 (35 %) when an accurate droplet number concentration could be prescribed to the algorithm.

Line 33: Can you quantify what you mean by large errors?

By this we mean a relatively high false alarm ratio and undetected events. The critical success index was shown by Philip et al. (2016) to be less than 0.4 for a winter season of forecasts in Paris by the operational model AROME.

Changed to: Despite the development of high-resolution numerical weather prediction (NWP) models, the forecast skill of these models is still lacking demonstrated by high rate of false alarm and undetected events in the case of the AROME model (Philip et al 2016).

Line 152: What is the source of the mask that defines the type of hydrometeor? A description or reference should be provided.

Added: The mask detects the melting layer from the radar reflectivity and Doppler velocity gradients. For the liquid section, rain, drizzle and cloud are defined from the Doppler velocity (Jorquera and Delanoë, 2020).

Line 702: "was" rather than "were".

**Changed as suggested**

**Bibliography**

Baumgardner, D., Abel, S.J., Axisa, D., Cotton, R., Crosier, J., Field, P., Gurganus, C., Heymsfield, A., Korolev, A., Kraemer, M. and Lawson, P., 2017. Cloud ice properties: In situ measurement challenges. *Meteorological monographs*, 58, pp.9-1.

Baumgardner, D. and Spowart, M., 1990. Evaluation of the Forward Scattering Spectrometer Probe. Part III: Time response and laser inhomogeneity limitations. *Journal of Atmospheric and Oceanic Technology*, 7(5), pp.666-672

Bell, A., Martinet, P., Caumont, O., Vié, B., Delanoë, J., Dupont, J.C. and Borderies, M., 2021. W-band radar observations for fog forecast improvement: an analysis of model and forward operator errors. *Atmospheric Measurement Techniques*, *14*(7), pp.4929-4946

Cimini, D., Westwater, E.R. and Gasiewski, A.J., 2009. Temperature and humidity profiling in the Arctic using ground-based millimeter-wave radiometry and 1DVAR. *IEEE Transactions on Geoscience and Remote Sensing*, *48*(3), pp.1381-1388.

Ebell, K., Löhnert, U., Crewell, S. and Turner, D.D., 2010. On characterizing the error in a remotely sensed liquid water content profile. *Atmospheric research*, *98*(1), pp.57-68.

Fabry, F., 2001, July. Using radars as radiometers: Promises and pitfalls. In *Preprints, 30th Int. Conf. on Radar Meteorology, Munich, Germany, Amer. Meteor. Soc* (Vol. 197, p. 198).

Löhnert, U., Crewell, S. and Simmer, C., 2004. An integrated approach toward retrieving physically consistent profiles of temperature, humidity, and cloud liquid water. *Journal of Applied Meteorology and Climatology*, *43*(9), pp.1295-1307.

Löhnert, U. and Maier, O., 2012. Operational profiling of temperature using ground-based microwave radiometry at Payerne: Prospects and challenges. *Atmospheric Measurement Techniques*, 5(5), pp.1121-1134.

Philip, A., Bergot, T., Bouteloup, Y. and Bouyssel, F., 2016. The impact of vertical resolution on fog forecasting in the kilometric-scale model arome: a case study and statistics. *Weather and Forecasting*, *31*(5), pp.1655-1671

Russchenberg, H., Crewell, S., Loehnert, U., Quante, M., Meywerk, J., Baltink, H.K. and Krasnov, O., 2004, September. Radar observations of stratocumulus compared with in situ aircraft data and simulations. In *Proc. ERAD* (pp. 296-300)

Tridon, F., Battaglia, A. and Kneifel, S., 2020. Estimating total attenuation using Rayleigh targets at cloud top: applications in multilayer and mixed-phase clouds observed by ground-based multifrequency radars. *Atmospheric Measurement Techniques*, *13*(9), pp.5065-5085.

Wendisch, M., Keil, A. and Korolev, A.V., 1996. FSSP characterization with monodisperse water droplets. Journal of Atmospheric and Oceanic Technology, 13(6), pp.1152-1165